# Global mapping of CARM1 substrates defines enzyme specificity and substrate recognition

Evgenia Shishkova[1,*], Hao Zeng[2,*,†], Fabao Liu[2], Nicholas W. Kwiecien[3], Alexander S. Hebert[3], Joshua J. Coon[1,4] & Wei Xu[2]

Protein arginine methyltransferases (PRMTs) introduce arginine methylation, a post-translational modification with the increasingly eminent role in normal physiology and disease. PRMT4 or coactivator-associated arginine methyltransferase 1 (CARM1) is a propitious target for cancer therapy; however, few CARM1 substrates are known, and its mechanism of substrate recognition is poorly understood. Here we employed a quantitative mass spectrometry approach to globally profile CARM1 substrates in breast cancer cell lines. We identified >130 CARM1 protein substrates and validated in vitro >90% of sites they encompass. Bioinformatics analyses reveal enrichment of proline-containing motifs, in which both methylation sites and their proximal sequences are frequently targeted by somatic mutations in cancer. Finally, we demonstrate that the N-terminus of CARM1 is involved in substrate recognition and nearly indispensable for substrate methylation. We propose that development of CARM1-specific inhibitors should focus on its N-terminus and predict that other PRMTs may employ similar mechanism for substrate recognition.

[1] The Department of Biomolecular Chemistry, University of Wisconsin – Madison, Madison, Wisconsin 53705, USA. [2] McArdle Laboratory for Cancer Research, University of Wisconsin – Madison, Madison, Wisconsin 53705, USA. [3] The Genome Center of Wisconsin, University of Wisconsin – Madison, Madison, Wisconsin 53705, USA. [4] The Department of Chemistry, University of Wisconsin – Madison, Madison, Wisconsin 53705, USA. * These authors contributed equally to this work. † Present address: Chemical Biology and Therapeutics, Novartis Institutes for Biomedical Research, Cambridge, Massachuetts 02139, USA. Correspondence and requests for materials should be addressed to J.J.C. (email: jcoon@chem.wisc.edu) or to W.X. (email: wxu@oncology.wisc.edu).

Protein arginine methylation is an abundant post-translational modification (PTM), catalysed by nine mammalian arginine methyltransferases (PRMTs)[1]. A recent study reveals that ~7% of all arginines are methylated that is comparable to 9% of serine residues being phosphorylated and 7% of lysine residues being ubiquitinated[2]. Via transferring a methyl group from S-adenosyl-L-methionine (SAM) to the side chain of arginine residues, PRMTs catalyse formation of three final product types: $\omega$-$N^G$-monomethylated arginine by PRMT7, $\omega$-$N^G,N^G$-asymmetric dimethylarginine (ADMA) by Type I PRMTs, and $\omega$-$N^G,N'^G$-symmetric dimethylarginine by Type II PRMTs. While arginine methylation does not alter amino-acid charge, it does increase its bulkiness and hydrophobicity[3]. This could affect protein–protein and protein–nucleic acid interactions[4], as a consequence, impacting myriad biological pathways[1,2], including transcription, RNA metabolism, DNA repair, among others. Aberrant expression and/or enzymatic activity of multiple PRMTs are associated with human cancers[5]; however, the functional significance of arginine methylation in oncogenic processes is poorly understood. This problem is further confounded by a lack of known cancer-relevant PRMT substrates[6]. Similarly, the absence of the full-length enzyme structure co-crystallized with known substrates hampers the effective design of PRMT-specific inhibitors[7].

Here we focus on PRMT4, also known as coactivator-associated arginine methyltransferase 1 (CARM1), an indispensable enzyme in mammals[8]. Both CARM1 knockout (KO) and enzyme-inactive CARM1 knock-in mice die at birth and display identical developmental defects (that is, defects in T-cell development and adipocyte differentiation)[8,9], underscoring the essentiality of CARM1's enzymatic function in physiology. Emerging evidence supports the pathological role of CARM1 in human disease, particularly in cancer. CARM1 is overexpressed in a variety of cancer types[10–12], and its higher expression correlates with poor prognosis[13,14]. Like several other PRMTs, CARM1 functions as both a coactivator and a methyltransferase[1]. CARM1 interacts with a plethora of transcription factors (for example, p53, oestrogen receptor and E2F1), modulating gene expression and contributing to cancer progression[15–17]. Besides histone H3 (ref. 18), CARM1 methylates non-histone substrates, likewise driving key oncogenic processes[5,6]. For example, CARM1-mediated methylation of NCOA3 (ref. 19) and BAF155 (ref. 20) promotes cancer progression and metastasis. Dissimilarly to PRMT1 and other PRMTs[21], where a well-characterized glycine and arginine-rich (GAR) methylation motif facilitates prediction of hundreds of protein substrates[22], in vivo substrate recognition motif(s) of CARM1 remain to be defined with fewer than 20 substrates known to date[5,6].

Given the prominence of CARM1 in oncogenesis, tremendous effort has been expended to design CARM1-specific inhibitors[7,23]. However, no inhibitor has been shown to be effective in animal models. This is in large due to the lack of structural insight into the physical basis of substrate recognition by CARM1, like by all PRMTs in general[24]. Members of the PRMT family share highly conserved central catalytic domain, but the primary sequences of N- and C-termini vary drastically. The catalytic core of CARM1, co-crystallized with several substrates, exhibits folding similar to that of other Type I PRMTs—the dimer form that accommodates SAM and peptide sequences[25]. The N- and C-termini of CARM1, however, appear disordered, and the structure of the full-length CARM1 has not been solved to date[25–27]. Intriguingly, when expressed alone, the N-terminal domain of CARM1 can be crystallized and displays a fold highly similar to a family of Drosophila-enabled/vasodilator-stimulated phosphoprotein homology 1 (EVH1) domain, a member of 'pleckstrin homology' (PH) domain superfamily[26]. Typically, EVH1 domains recognize and bind proline-rich sequences[28]; however, the precise function of the N-terminal domain in CARM1 remains enigmatic[24].

In this study we employed high-resolution mass spectrometry (MS) and the newly developed ADMA antibody[29] to globally profile CARM1 substrates in two human breast cancer cell lines. We distinguished a decrease in substrate methylation from the total protein changes induced by CARM1 loss, leading to the precise mapping of arginine methylation events regulated by CARM1 in vivo. Altogether, we identified over 300 CARM1-dependent ADMA sites, encompassed by ~130 novel bona fide CARM1 protein substrates. Many of these substrates have cancer-relevant functions and thus are possible mediators of CARM1's oncogenic potential. In vitro methylation array confirmed the ability of CARM1 to methylate over 90% of the tested sequences. Further, informatic analysis revealed the presence of proline-rich motifs nearby CARM1 methylation sites. Both CARM1-methylated arginines and the surrounding recognition sequences were frequently targeted by somatic mutations in cancer, likely inducing reduction or complete abolishment of methylation by CARM1. Finally, we discovered that the N-terminal EVH1 domain of CARM1 is necessary and sufficient for substrate recognition and is required for methylation of most CARM1 substrates. This finding opens new routes in the design of CARM1-specific inhibitors and warrants functional investigation of the N-terminal domains of other PRMT family members in substrate recognition.

## Results

**Global profiling of CARM1 substrates in breast cancer cells.** Our previous studies identified two CARM1 substrates, BAF155 (ref. 20) and MED12 (ref. 30), by individually selecting the differentially precipitated proteins by ADMA pan-specific antibody in CARM1 KO MCF7 cells and the parental cells with endogenous CARM1 expression. Here we aimed to globally profile CARM1 substrates in vivo. To do this, we immunoprecipitated ADMA-containing peptides using ADMA antibodies in two parental and CARM1 KO paired cell lines, MCF7 and MDA-MB-231, and mapped the sites of arginine methylation using nano-liquid chromatography tandem mass spectrometry (nanoLC-MS/MS) in biological triplicates (Fig. 1a). Our experimental design capitalized on the multiplexing capabilities of tandem mass tags (TMT)[31], a technique that allowed us to simultaneously enrich and analyse ADMA-modified peptides from wild type and KO samples, assuring a complete overlap of peptide identifications between the two groups. This approach evades the common pitfall of PTM analysis—poor enrichment reproducibility[32]. Note the process of correlating tandem mass spectra to peptide sequences can be confounded in cases where the spectra originate from methylated peptides[33]. To counter this potential issue, we sequenced all MS/MS spectra corresponding to ADMA-containing peptides (deposited in PRIDE[34] accession #PXD005871), confirming their accurate identification and ensuring correct localization of methylation sites. Neutral loss of dimethylamine, characteristic of ADMA peptides and not their isobaric symmetrically methylated counterparts[35], was systematically observed in the annotated spectra (Fig. 1b). Consideration of this loss improved achieved sequence coverage by ~4% ($71.5 \pm 21.7\%$ and $75.2 \pm 21.2\%$ excluding and including the neutral loss, respectively), further fostering our confidence in identified peptides. Additionally, good reproducibility was observed among biological replicas across all experiments with median $R^2$ of 0.98 for both modified and unmodified peptides (Supplementary Fig. 1a).

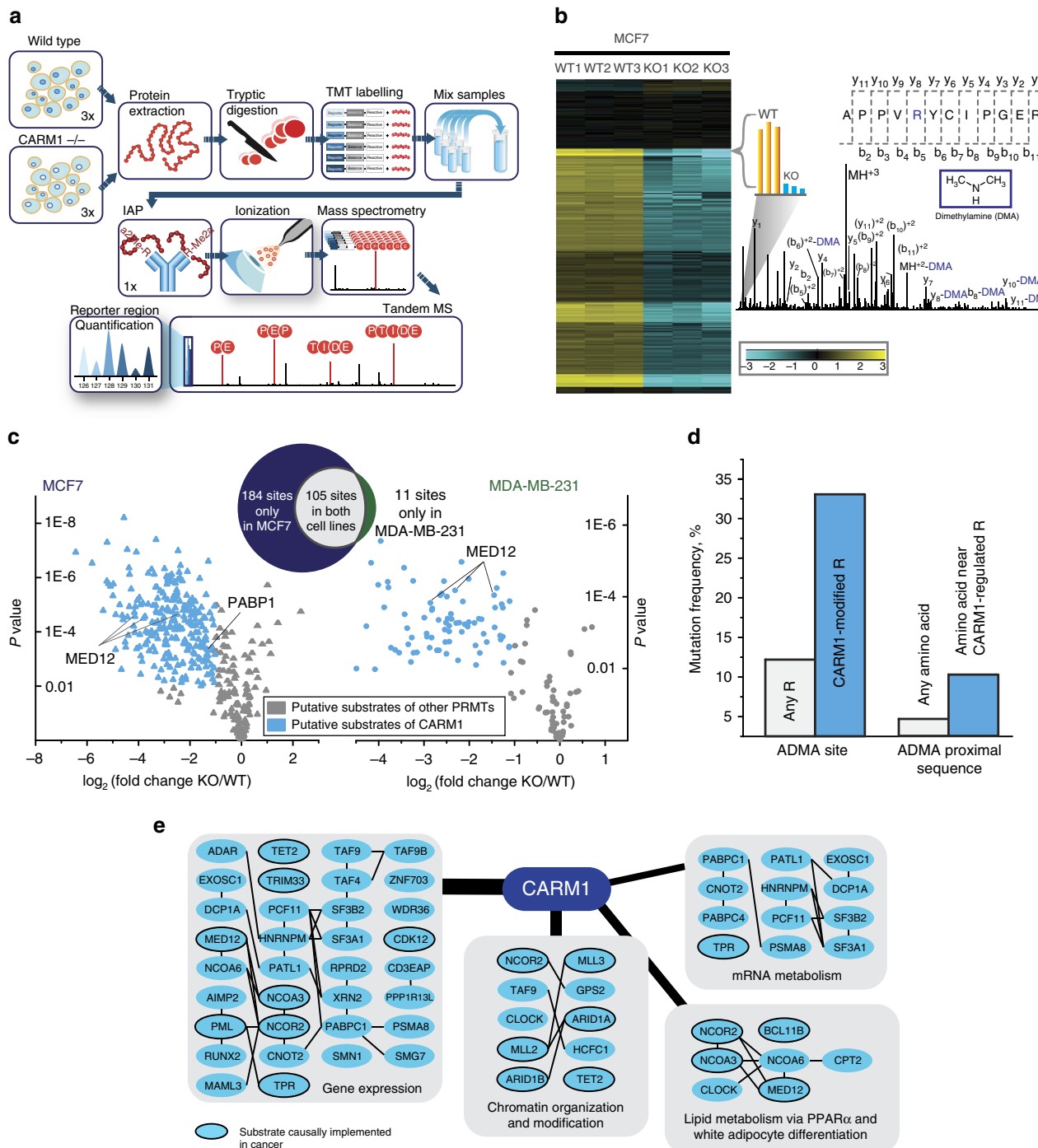

**Figure 1 | Global profiling of CARM1 substrates using quantitative mass spectrometry.** (**a**) Experimental workflow for global detection of CARM1 substrates. Enrichment of ADMA—containing tryptic peptides from three biological replicas of parental and CARM1 KO cell lines—was followed by quantitative mass spectrometry using TMT. (**b**) Identification and quantification of ADMA-containing peptides and sites. The heat map displays hierarchical clustering using Pearson correlation of mean normalized $\log_2$ transformed intensities of ADMA containing peptides in three biological replicas of parental MCF7 and CARM1 KO cells. A TMT reporter region of the spectrum is magnified as an example. The spectrum also illustrates the neutral loss of DMA, characteristic of asymmetric but not symmetric dimethylarginine modification that was used for peptide identification and ADMA site mapping. (**c**) Identification of putative CARM1 substrates from two breast cancer cell lines using three biological replicas of each sample. Volcano plots illustrate changes in ADMA peptide abundances in MCF7 (left) and MDA-MB-231 cells (right). Greater than twofold reduction on CARM1 loss and maximum $P$ value of 0.01 (two-tailed Student's $t$-test) were set as the threshold criteria for putative CARM1 substrates. The known CARM1 protein substrates, MED12 and PABP1, were identified among >130 previously unreported ones. (**d**) Rates of somatic mutations at and nearby CARM1 methylation sites in human cancers. Frequencies of non-synonymous single nucleotide variants at or in the proximity of CARM1-regulated ADMA sites ($\pm 5$ nucleotides) were over twofold higher than those of non-modified arginines or a randomly selected residue (Fisher's exact test $P$ values of 5.6e-31 and 7.8e-24, respectively), according to the COSMIC. (**e**) Substrate interaction diagram (STRING 10.0) featuring four biological pathways (Reactome 2016) strongly enriched for the presence of putative CARM1 substrates (combined score >5). Thickness of the lines radiating from CARM1 correlates to the pathway enrichment score (Supplementary Table 3). A black frame around a substrate indicates its causal implementation in cancer (COSMIC).

In both MCF7 and MDA-MB-231 cells, we observed pronounced reduction in the levels of ADMA-containing peptides on deletion of CARM1 (Fig. 1b and Supplementary Fig. 1b, respectively; Supplementary Data 1). Specifically, in CARM1 KO MCF7 cells, over 50% of the detected modified peptides, encompassing nearly 300 unique ADMA sites, decreased in abundance by greater than twofold, as compared to the parental cells. In both cell lines such drastic reduction in abundance uniquely affected modified peptides, as less than 1% of unmodified peptides exhibited changes comparable in magnitude (Supplementary Fig. 1c).

Because ADMA could be deposited by all Type I PRMTs[1], we segregated putative CARM1 substrates from substrates of other PRMTs by evaluating fold changes in response to CARM1 loss (Fig. 1c). The abundance of an ADMA site on PABP1, a known CARM1 substrate in MCF7 cells[20,36], decreased by slightly over twofold. Thus, to stratify putative CARM1 substrates from those of other PRMTs, we selected ADMA sites that were reduced by twofold or more in the CARM1 KO cells ($P$ value < 0.01; two-tailed Student's $t$-test). In both cell lines this threshold correctly categorized three MED12 peptides, harbouring known CARM1 methylation sites[30,37], as CARM1 substrates. Fewer ADMA-containing peptides were detected in MDA-MB-231 cells as compared to MCF7 cells, likely due to differential substrate expression and/or CARM1 activity between two cell lines, as well as variable enrichment success. However, a considerable overlap was observed between sites identified in MCF7 and MDA-MB-231, even though some cell-type-specific substrates were identified.

Due to CARM1's function as a transcription coactivator, loss of CARM1 is expected to alter expression of some proteins[8,9]. To ensure that the observed reduction in the levels of ADMA-containing peptides was correctly attributed to the reduction in arginine methylation, rather than resulted from the underlying decrease in total abundance of the respective proteins, we quantified and compared protein levels of putative CARM1 substrates in parental and CARM1 KO cells (Supplementary Fig. 1d and Supplementary Data 2). Across both cell lines, only four substrates exhibited decreased abundance (~twofold) upon CARM1 loss. We also noted that the levels of two other substrates exhibited moderate increases in abundance (less than threefold). By and large, however, nearly all the substrates showed no detectable modulation. We conclude that the observed reduction in ADMA peptide abundances were due to the loss of CARM1 catalysed methylation on arginine, rather than changes in protein abundance.

Finally, we examined levels of other PRMTs (PRMT1, 2, 5, 6 and 8) to ensure that changes in their abundance did not produce the observed reduction in ADMA-containing peptides. Together, mass spectrometric measurements and western blot analyses (Supplementary Data 2 and Supplementary Fig. 1d, respectively) confirmed that in both cell lines, levels of all PRMTs, but PRMT1 in MCF7 cells, were unchanged on CARM1 deletion. Note, PRMT8 was not detected by both MS and western blotting (data not shown), likely due to its brain-specific expression[1]. The abundance of PRMT1 in MCF7 cells was slightly reduced by ~2.5-fold, and the implications of this change are discussed later.

**Many CARM1 substrates have known oncogenic functions.** Overall, we identified over 300 unique ADMA sites on 138 different proteins that may be modified by CARM1 *in vivo*; over 90% of them have not been previously reported as substrates of the enzyme. According to the Catalogue of Somatic Mutations in Cancer (COSMIC[38]), frequencies of non-synonymous single nucleotide variants at, and in the proximity of CARM1-regulated ADMA sites ( ± 5 nucleotides), were over twofold higher as compared to the whole human proteome (Fig. 1d). This significant enrichment (Fisher's exact test $P$ values of 5.6e–31 and 7.8e–24) indicates that CARM1 methylation sites and recognition sequences are vulnerable to mutation in cancers and may provide a functional advantage. COSMIC analysis also reveals that 17 of the putative CARM1 substrates are causally implemented in cancer pathogenesis (Fig. 1e and Supplementary Table 1). Further, pathway enrichment analysis[39] and protein–protein interaction[40] analyses illuminated the critical role of these proteins in diverse biological pathways that are commonly deregulated in cancers (Fig. 1e and Supplementary Table 2). Unlike PRMT1 with most of its substrates involved in RNA processing and surveillance[2], pronouncedly, 34 putative CARM1 substrates are involved in control of gene expression. Ten substrates possess functions in chromatin organization and enzymatic modifications. Various mRNA-processing pathways, such as mRNA stability, decay and splicing, also appeared over-represented. Interestingly, lipid metabolism via PPAR-alpha pathway and transcription regulation of white adipocyte differentiation were also identified, consistently with the reported functions of CARM1 in adipogenesis[41]. The functions of CARM1 substrates in HOX and TP53 signalling pathways were also mildly enriched. Together, identification of many cancer-relevant substrates across diverse biological pathways and the elevated mutation rate at, or near, CARM1-methylated arginines underscore the significance of substrate methylation by CARM1 in oncogenic processes.

**Detection of proline-rich motifs among CARM1 substrates.** With this extensive collection of CARM1 substrates, we sought to identify a consensus substrate recognition motif. Note no such motif confirmed in living cells presently exists[1,5,6]. A comparison of amino-acid frequencies in the human proteome and in the 11 amino-acid sequences centring on the putative CARM1 methylation sites revealed an over threefold enrichment of proline residues in the vicinity of the ADMA-bearing arginines (Fig. 2a). These results confirm the enrichment of proline among CARM1 methylation substrates previously inferred by Cheng *et al.*[42]; however, they dispute the significance of methionine and glycine residues in the so-called proline-, glycine- and methionine-rich (PGM) motif as the hallmark of CARM1 substrate recognition. Specifically, methionine was detected at frequencies typical for the human proteome. The abundance of glycine residues appeared slightly elevated, but the difference was not significant ($P$ value of 0.13).

Using the MotifX algorithm[43], we extracted an assortment of proline-containing motifs from the CARM1 methylation sequences (Fig. 2b and Supplementary Table 3). The resultant motifs feature proline residues in various positions up to five amino acids away from the central arginine. Several of these motifs are in good agreement with the termed 'proline-rich arginine methylation' motifs, previously reported in a global arginine methylation study[44]. As expected, the GAR motif and highly similar sequences, typically methylated by other PRMTs but not CARM1 (ref. 1), were detected in the vicinity of the methylation sites whose abundance was unaffected by the loss of CARM1 (Supplementary Fig. 2a and Supplementary Table 4). As our methodology did not permit direct measurements of abundance changes on individual ADMA sites in the context of multiply dimethylated peptides, we performed control analyses by separately extracting motifs from singly modified peptides. The detected motifs (Supplementary Fig. 2b,c) were nearly identical to a subset of motifs on Fig. 2b and Supplementary Fig. 2a, therefore further corroborating their enrichment.

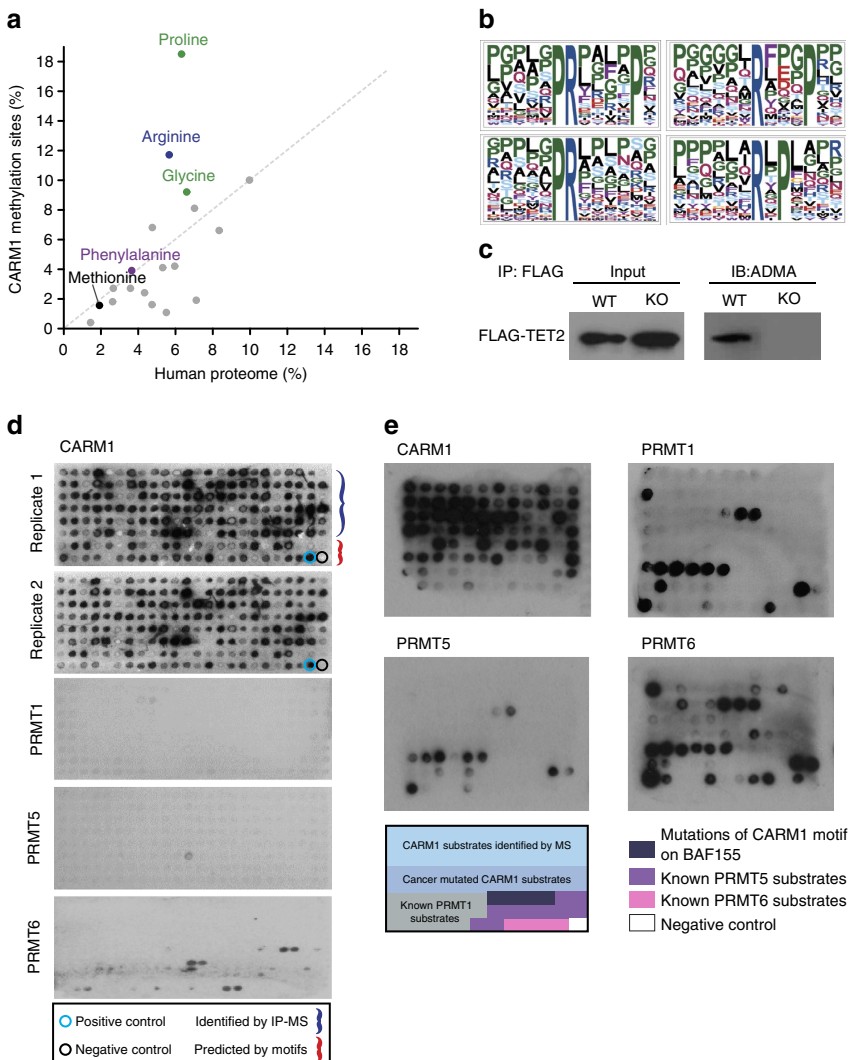

**Figure 2 | Discovery of proline-rich CARM1 methylation motifs and high-throughput validation of the detected substrates using peptide arrays.**
(**a**) Amino-acid abundance analysis of CARM1 methylation sites. Frequencies of amino acids in the vicinity of CARM1-regulated methylation sites ( ± 5 nucleotides) were compared to those in the human proteome, revealing pronounced enrichment of proline residues. Glycine, methionine and phenylalanine residues were detected at frequencies typical of the human proteome (P value > 0.01; two-tailed Student's t-test). (**b**) Sequence logos of CARM1 methylation motifs. Various proline-containing motifs were enriched in the proximity of the detected CARM1 methylation sites. (**c**) Western blot with ADMA antibodies of FLAG-tagged TET2 in wild type and CARM1 KO HEK293T cells showing the reduction of TET2 dimethylation in CARM1 KO. (**d**) Autoradiography of 192-spot peptide arrays following the *in vitro* methylation assay using [3]H-labelled SAM and indicated purified mammalian PRMTs. The peptide arrays consisted of ∼15 amino-acid sequences centred on substrate arginines. Designated by the blue bracket are substrates of CARM1 identified by IP-MS; the red bracket indicates substrates predicted based on the extracted motifs. Positive control (BAF155 peptide) is denoted with a blue circle; negative control (no peptide) is indicated with a black circle. (**e**) Autoradiography of 96-spot peptide arrays following the *in vitro* methylation assay using [3]H-labelled SAM and indicated purified mammalian PRMTs. The design of peptide array is depicted at the bottom. Light blue colour designates the location of novel, cancer-related substrates of CARM1 identified by IP-MS; blue—additional substrates and their naturally occurring mutations in human cancer; dark-blue—point mutations of the recognition sequence adjacent to the CARM1 methylation site on BAF155; grey—known substrates of PRTM1; purple—known substrates of PRMT5; pink—known substrates of PRMT6; white—negative control (no peptide).

**High-throughput validation of CARM1 substrates.** TET2, methylcytosine dioxygenase with a prominent role in DNA demethylation frequently mutated in various human cancers[45], was identified among novel CARM1 substrates. To assure its correct identification, we probed FLAG-tagged TET2 with ADMA antibody in wild type and CARM1 KO HEK293T cells (Fig. 2c). The western blot revealed complete disappearance of ADMA modification on TET2 in the absence of CARM1, confirming TET2 as a bona fide CARM1 substrate.

To validate the putative CARM1 substrates in a high-throughput fashion, we designed a peptide array containing nearly 200 peptide sequences of 15 amino acids in length

(Supplementary Data 3); three-quarters of the included peptides were centred on the putative CARM1 methylation sites detected by immunoprecipitation combined with MS (IP-MS). To assess the predictive power of our findings, we examined the human proteome in search of proteins harbouring the identified proline-rich motifs and selected 42 additional sequences. The peptide arrays were *in vitro* methylated by purified recombinant PRMTs (Supplementary Fig. 2d) in the presence of[3]H-SAM and visualized by autoradiography. The experiment confirmed that over 90% of the putative substrate sequences could be methylated by CARM1 *in vitro* (Fig. 2d). Notably, ∼85% of the CARM1 substrates predicted based on the extracted proline-rich motifs

were also methylated, showcasing the extrapolative power of the identified substrate recognition motifs. In contrast to CARM1, PRMT1 and 5 failed to methylate these peptides, although PRMT6 could weakly methylate a few of the tested sequences.

Having corroborated the ability of CARM1 to methylate most of the detected substrates, we designed an additional 96-spot peptide array (Fig. 2e and Supplementary Data 3) with two purposes in mind: (1) to examine the significance of surrounding amino acids on CARM1-directed arginine dimethylation, particularly those mutated in human cancers; and (2) to measure PRMT specificity towards respective substrates. The selected sequences encompassed a mixture herein identified and predicted CARM1 substrates (in light blue) and representative cancer-relevant substrates, accompanied by the sequences harbouring mutations detected in human patients[38] (in blue). We also arbitrarily mutated the recognition sequence surrounding R1064 on BAF155 (ref. 20)—the known CARM1 methylation site—to assess the significance of surrounding amino-acid residues to CARM1 activity (four sequence in dark blue). The array also featured 17 known substrates of PRMT1 (in grey), 10 substrates of PRMT5 (in purple) and 4 substrates of PRMT6 (in pink) as positive controls for these enzymes.

The results of the *in vitro* methylation assay using purified CARM1 visualized via autoradiography (Fig. 2e, top left array) confirmed its ability to methylate over 90% of the tested putative substrates. The comparison between the wild-type sequences and their mutated counterparts revealed pronounced reduction to complete abrogation of CARM1's ability to methylate the substrates. For example, the naturally occurring mutations on TET2 (R680) G678D, ARID1A (R429) Y430C and P431Q, and MED12 (R1862) P1864Q significantly reduced methylation by CARM1, underscoring their probable functional significance in tumorigenesis. Similarly, mutagenesis of the BAF155 (R1064) surrounding sequence demonstrated that mutations of the proximal residues (P1063A, G1062A and P1066A) nearly abolished methylation by CARM1. Together, these findings clearly demonstrate that the residues nearby the methylated arginine play a critical role in substrate recognition/methylation.

Assays using purified PRMT1, 5 and 6 (Fig. 2e, as indicated) revealed that a few substrates could be methylated by both CARM1 and other PRMTs. Specifically, ~8, 5 and 19% of the tested CARM1 substrates were also methylated by PRMT1, PRMT5 or PRMT6, respectively. Concomitant control experiments with respective known substrates of these PRMTs validated their normal enzymatic activity; PRMT1 methylated 9 out of 17 tested substrates, PRMT5–2 out 10, and PRMT6–3 out 4. Overall, these data demonstrate that although some overlap in substrate preference occurred among PRMTs, CARM1 was prevalently selective towards the identified peptide sequences designated as CARM1 substrates by the LC-MS/MS experiments.

**The N-terminus of CARM1 in substrate recognition**. The structure of the N-terminal domain of CARM1 (residues 28–140) resembles those of the EVH1 domains, a subfamily of PH domains that bind proline-rich sequences[26]. With our discovery that CARM1 substrates contain proline-rich motifs, we posited that the protein's N-terminal domain may be required for substrate recognition. To test this hypothesis, we generated three CARM1 constructs: one full-length (CARM1 FL) and two N-terminal truncations CARM1 28–608 and CARM1 140–608 (Fig. 3a). CARM1 140–608 eliminates the entire EVH1 domain, while CARM 28–608 was included as a control, since the first 28 residues at the N-terminus of CARM1 do not participate in the formation of the EVH1 domain[26]. We transiently transfected the three FLAG-tagged CARM1 fusion constructs in HEK293T

CARM1 KO cells. CARM1 was immunoprecipitated from all three cell lines using anti-FLAG M2 resin, and four known CARM1 substrates/interactors (that is, BAF155 (ref. 20), MED12 (ref. 30), PABP1 (ref. 36) and NCOA3 (ref. 19)) and newly identified TET2 were detected by western blotting. Figure 3b displays that all five proteins were co-precipitated with CARM1 FL and CARM1 28–608; however, substantially lower amounts of them were detected with CARM1 140–608. To globally assess the magnitude of reduction in CARM1–substrate interactions induced by the loss of the EVH1 domain, we visualized overall protein changes in CARM1 immunoprecipitates by western blotting using the ADMA antibody (Fig. 3c). The experiment revealed global decrease in abundance of CARM1-interacting proteins with CARM1 140–608, as compared with the CARM1 FL and CARM1 28–608.

These results support our hypothesis that CARM1 may employ its N-terminus for substrate recognition. To further test this, we performed fluorescence polarization (FP) assays using purified, recombinant 6xHis-tagged CARM1 EVH1 domain and fluorescein-labelled BAF155 peptide harbouring the CARM1 methylation site (Fig. 3d,e). The assay detected dose-dependent increase in polarization signals as concentration of the CARM1 EVH1 domain increased, whereas the control experiment using bovine serum albumin (BSA) generated no change in signal. A similar result was obtained when the His-tag was replaced with glutathione S-transferase (GST), excluding possible effects of the fusion tag binding (Supplementary Fig. 3). Together, these findings strongly indicate that the N-terminal EVH1 domain of CARM1 can directly interact with CARM1 substrates so that its deletion broadly impacts substrate recognition by the enzyme.

**The N-terminal domain of CARM1 in substrate methylation**. Having established that the N-terminal domain of CARM1 mediates substrate recognition, we next examined its prerequisite for substrate methylation. First, we expressed and purified recombinant CARM1 proteins, including CARM1 FL, CARM1 28–608 and CARM1 140–608, for *in vitro* methylation assays. Autoradiography results showed that CARM1 FL and CARM1 28–608, but not CARM1 140–608, methylated recombinant full-length BAF155 protein (Fig. 4a). Similar observations were made when GST fused MED12, BAF155 and PABP1 peptides encompassing CARM1 methylation sites were used as substrates in *in vitro* methylation assays (Supplementary Fig. 4a,b,c). The greatly reduced ability of CARM1 140–608 to methylate protein substrates, as compared with CARM1 FL and CARM1 28–608, were not restricted to a few substrates. When total cell lysates derived from MCF7 CARM1 KO cells were *in vitro* methylated, we observed substantial reduction of total protein methylation in assays employing CARM1 140–608 (Fig. 4b). These results demonstrate that the N-terminal EVH1 domain of CARM1 is required for substrate methylation *in vitro*.

Next, we transiently transfected FL and the two aforementioned CARM1 truncations in HEK293T CARM1 KO cells and measured substrate methylation by western blotting with ADMA antibodies. Not only methylation of endogenous BAF155 and MED12 was restored in the presence of CARM1 FL and CARM1 28–608 (Supplementary Fig. 4d), but the total levels of ADMA modification were recovered to the level of parental cells (Supplementary Fig. 4e). In contrast, transient expression of CARM1 140–608 failed to restore endogenous protein methylation in HEK293T CARM1 KO cells. These data suggest that the N-terminal EVH1 domain is indispensable for CARM1 substrate recognition and methylation both *in vitro* and in cells transiently expressing CARM1.

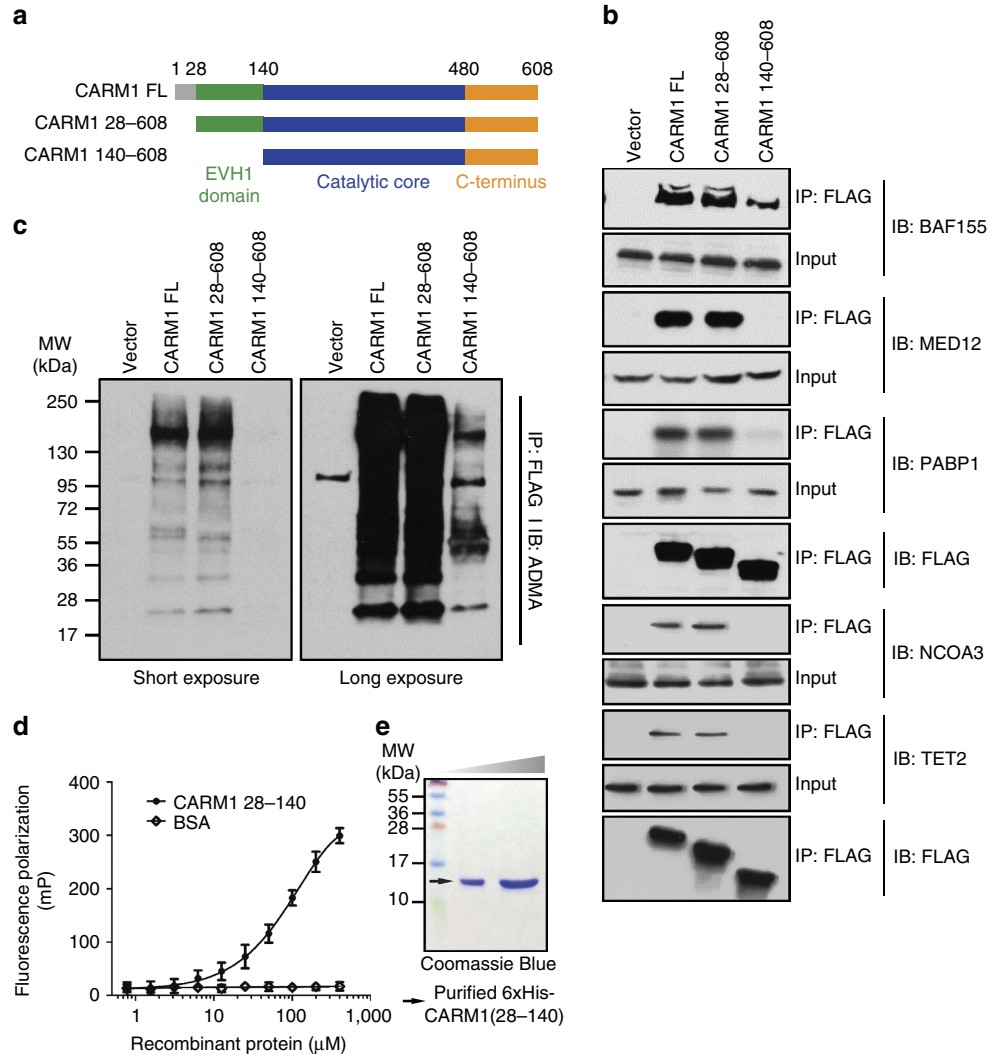

**Figure 3 | Requirement of the N-terminal domain for substrate recognition by CARM1.** (**a**) Schematic diagram of FL and N-terminal truncated CARM1 derivatives. CARM1 FL protein contained 608 residues. CARM1 28–608 lacks the first, unstructured 28 residues denoted in grey. CARM1 140–608 lacks the first 140 residues encompassing the EVH1 domain denoted in green. (**b**) Western blot analyses of co-immunoprecipitated BAF155, MED12, PABP1, NCOA3 and TET2 with FLAG-tagged CARM1, transiently transfected into HEK293T CARM1 KO cells. CARM1 was immunoprecipiated with the anti-FLAG antibody, and the presence of BAF155, MED12, PABP1, NCOA3 and TET2 in the immunoprecipitates was detected with western blots using the respective antibodies. The loading controls are depicted below the corresponding western blot results, separately for BAF155, MED12 and PABP1, and the other two proteins. In all cases the amount of co-precipitated protein was strongly reduced in cell lines expressing N-terminus truncated CARM1 140–608.
(**c**) Western blot analyses of total ADMA-containing proteins co-precipitated with CARM1. The FLAG-tagged CARM1 immunoprecipitates in **b** were probed with ADMA antibodies. The strong reduction in the levels of ADMA—containing proteins in cells expressing CARM1 140–680 was evident on both short (left) and long exposure (right). The corresponding loading control was shared between the experiments in **b** (BAF155, MED12 and PABP1) and **c** and is depicted in **b** labelled with IB: FLAG. (**d**) FP assay using purified recombinant 6xHis-CARM1 28–140 and fluorescein-labelled BAF155 peptide. Pronounced increase in FP was observed at high concentrations of recombinant CARM1, but not with the BSA control, demonstrating that the EVH1 domain of CARM1 directly interacts with the enzyme's substrate at low affinity. (**e**) Coomassie Blue staining of highly purified recombinant 6xHis-CARM1 28–140 used in the FP assay (**d**).

We next sought to assess the impact of CARM1 N-terminal deletion in cells stably expressing mutant enzymes. We stably expressed GFP (control) and the previously described CARM1 constructs in MCF7 CARM1 KO cells and observed impaired *in vivo* methylation of BAF155 in CARM1 140–608 expressing, MCF7 CARM1 KO cells (Fig. 4c). Western blots of total cell lysates using ADMA antibody similarly revealed that GFP and CARM1 140–608 expressing cells exhibited drastically decreased levels of cellular protein methylation, as compared with CARM1 FL and CARM1 28–608 expressing MCF7 cells (Fig. 4d). To quantify the affected ADMA-containing peptides in MCF7 CARM1 KO and CARM1 140–608 expressing cells, we employed our quantitative MS method (Fig. 1a) using CARM1 28–608 as a reference for normalization (Supplementary Data 4). This experiment revealed a massive reduction in abundances of ADMA-containing peptides in the CARM1 140–608 expressing and complete CARM1 KO cells (Fig. 4e). The magnitude of reduction in CARM1 140–608 cells, however, was slightly less than that in the CARM1 KO cells. Figure 4f directly compares $\log_2$ transformed intensities of ADMA-containing peptides that met the criteria for CARM1 substrates in both cell types, that is, exhibited greater than twofold reduction with maximum $P$ value of 0.01. The line of best fit suggests the overall good agreement between measured peptide intensities ($R^2$ of 0.82), but the

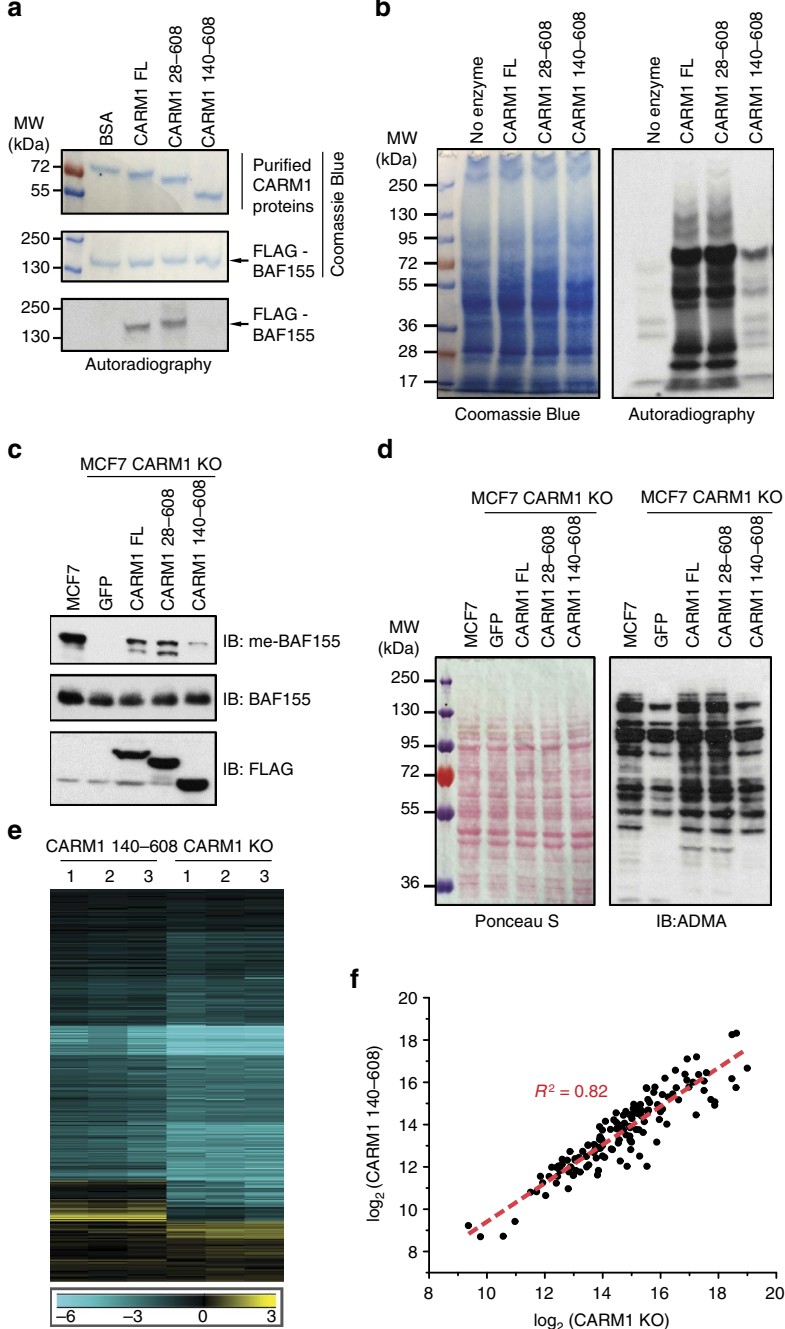

**Figure 4 | *In vitro* and *in vivo* requirement of the N-terminal domain for substrate methylation by CARM1.** (**a**) *In vitro* methylation assays using recombinant CARM1 proteins and [3]H-SAM to test methylation of FLAG-BAF155 protein. Coomassie Brilliant Blue staining of purified BSA (control), FL CARM1, the N-terminally truncated CARM1 28–608 and 140–608, and FLAG-BAF155 are shown in the top two panels. Radioactive labelling of BAF155 is visualized using autoradiography after *in vitro* methylation assays. (**b**) *In vitro* methylation assay using recombinant CARM1 proteins, [3]H-SAM and total cell lysates derived from MCF7 CARM1 KO cells. Coomassie Brilliant Blue staining (left panel) and autoradiograph (right panel) of total cell lysates after *in vitro* methylation assays are shown. (**c**) Western blot analyses of *in vivo* methylated BAF155 in parental MCF7 or MCF7 CARM1 KO cells stably expressing GFP control, CARM1 FL, CARM1 28–608 or CARM1 140–608. Expression of CARM1 in total cell lysates was detected by anti-FLAG antibody (bottom panel) and the levels of me-BAF155 (top panel) and total BAF155 (middle panel) were detected using corresponding antibodies. (**d**) Western blot analyses of ADMA-containing proteins in the total cell lysates as described in **c**. While Ponceau S staining (left panel) confirms equal loading, Western blot (right panel) demonstrates the reduced levels of ADMA in GFP control and CARM1 140–608 expressing cells, as compared with those in parental cells and cells expressing CARM1 FL and CARM1 28–608. (**e**) Heat map displaying hierarchical clustering of $\log_2$ transformed ADMA peptide intensities in complete CARM1 KO MCF7 cells and CARM1 KO, CARM1 140–608 expressing MCF7 cells, using three biological replicas of each. The intensities were normalized to their average respective levels in CARM1 KO, CARM1 28–608 expressing MCF7 cells. The ADMA peptides were prepared and quantified via nanoLC-MS/MS analysis, as described in Fig. 1a. (**f**) Comparison of the abundance of ADMA-containing peptides in CARM1 KO MCF7 cells and CARM1 KO MCF7 cells expressing CARM1 140–608. A close Pearson correlation in the levels of ADMA-containing peptides was detected in two cell lines ($R^2$ of 0.82).

reduced slope of the line indicated that the abundance of methylated peptides in cells expressing CARM1 140–608 was slightly, but systematically, higher than in cells completely lacking the enzyme. As indicated by $R^2$ of 0.65, noticeable differences existed between the levels of ADMA-containing peptides in cells expressing the N-terminal truncation of CARM1 and CARM1 28–608 control (Supplementary Fig. 4f). These data verify that the N-terminal EVH1 domain of CARM1 is necessary for recognition and methylation of its many endogenous substrates, even though a small fraction of them could bypass this requirement.

Surprisingly, the abundance of a few ADMA-containing peptides (yellow cluster, Fig. 4e) increased in CARM1 KO and CARM1 140–608 cells, as compared with the reference CARM1 28–608 cells. This subpopulation of sites experienced increased levels of ADMA methylation when CARM1 was absent. We surmise that these ADMA-containing peptides are not bona fide CARM1 substrates but are substrates of other Type I PRMTs that are affected by CARM1. As our MS data and western blot results (Supplementary Fig. 1d,e) previously suggested PRMT1 was reduced by ~2.5-fold in CARM1 KO MCF7 cells. Substrate scavenging by other PRMTs on decrease in the levels of PRMT1 was recently reported by Dhar et al.[46], possibly accounting for the unexpected spike in abundance of some ADMA-containing peptides. Consistently with this suggestion, we detected several previously described PRMT1 substrates among the sites upregulated on CARM1 deletion and consequent reduction in PRMT1 (refs 1,5,6) (Supplementary Data 5). The change in PRMT1 level induced by CARM1 deletion was not observed in CARM1 140–680 cells (Supplementary Fig. 5a,b) and in MDA-MB-231 cells (Supplementary Fig. 1c), suggesting that CARM1 regulation of PRMT1 may be cell line specific and is a property of the FL enzyme.

## Discussion

Using quantitative mass spectrometry along with pan-specific ADMA antibodies, we comprehensively profiled substrates of CARM1, the prototype PRMT with the strongest link to oncogenesis[5,6,12], in two breast cancer cell lines. Specifically, we discovered in vivo over 300 CARM1-dependent arginine methylation events, corresponding to >130 novel CARM1 protein substrates, many of which have documented cancer-relevant functions. The large overlap in the identified substrates between two cell lines combined with high-throughput in vitro validation engenders a high confidence compendium of CARM1 substrates. Though our study expands on the known methylation sites of CARM1 by over 10-fold, doubtlessly additional substrates remain elusive. Several factors likely impact the efficiency of substrate discovery using the herein employed approach. First, extremely low-abundance substrates may fall below our limit of detection[32], and substrate abundances almost certainly fluctuate across cell lines (Fig. 1c). Second, some ADMA sites likely exist within sequences that are impenetrable on trypsin digestion. For example, the known CARM1 methylation site on BAF155, R1064 (ref. 20), is located in a region whose nearby 100 amino acids are devoid of lysine and arginine residues necessary for trypsin cleavage. As the result, BAF155 was not identified by the IP-MS experiments. We envision that future works using other cell lines and possibly different digestion proteases will likely expand on this foundational data set.

Building on our compendium of in vivo methylated substrates, we uncovered new CARM1 substrate recognition motifs (Fig. 2a,b). An earlier study suggested the importance of proline, glycine and methionine residues in the proximity of CARM1 methylation sites (that is, the PGM motif)[42]. Our findings confirm the enrichment of proline; however, they do not validate the proposed significance of either glycine or methionine residues in CARM1 recognition sequences. Further, several amino acids identified as components of the CARM1 methylation motif by Gayatri et al.[37] did not appear enriched or were selected against according to our results. For example, large aromatic amino acids, therein reported as the essential components of the CARM1 substrate recognition motif, were rarely found within our substrate sequences.

A unique combination of three facets distinguishes our work from these prior studies: (1) measurement of in vivo methylated substrates, (2) high-throughput validation of putative substrates in vitro and (3) successful in silico prediction of novel substrates. As such, we are confident that the revisions we offer to the existing knowledge of CARM1 recognition motifs are bona fide and will expedite further expansion of its substrate repertoire. For example, histone H3R17 and H3R26 are known CARM1 modification sites[18] whose flanking sequences match the 'PR' and 'RxxxP' motifs (where x designates any amino acid; Fig. 2b), respectively. On the contrary, H3R2, which does not contain the detected motifs, is a putative CARM1 methylation site that is, however, called into question by in vivo evidence[47,48]. Still, we acknowledge that consensus recognition motifs likely do not universally apply to all substrates, as our findings indicate that ~6% of the detected CARM1 methylation sites do not contain proline residues in their vicinity (±5 amino acids).

Building on our improved CARM1 methylation motifs, we established the function of the enzyme's N-terminal domain in substrate recognition. Toffer-Charlier et al.[26] had previously detailed structural similarity between the N-terminus of CARM1 and EVH1 domains that recognize and bind proline-rich sequences; however, based on this observation, the authors hypothesized that the domain may participate in protein–protein interactions, enabling the cofactor function of CARM1. On the contrary, our results unambiguously demonstrate that the N-terminal EVH1 domain of CARM1 is necessary for substrate recognition and methylation, aiding the enzymatic, not cofactor, activity of CARM1.

Interestingly, the N-terminus of CARM1 lacks conserved aromatic residues that are typically located at the binding interface of EVH1 domains and engender their ligand specificity. Suitably, the absence of the conserved phenylalanine residues is the key difference between the classical consensus-binding motifs of the EVH1 domains[28] and the CARM1 recognition motifs (Fig. 2b). The structural divergence and dissimilarities in recognition motifs clearly distinguish CARM1 from other proteins harbouring EVH1 domains, likely ensuring that CARM1 could outcompete the latter in specifically recognizing and subsequently methylating its substrates.

In the study by Toffer-Charlier et al.[26], the N-terminal domain of the nearly FL CARM1 (residues 28–507) appeared disordered and, therefore, was dubbed the wobbly PH domain. We speculate that substrate binding may be a prerequisite for formation of the N-terminal PH fold, and the resultant, structurally defined N-terminal domain may assist in capturing substrates and sandwiching them against the catalytic core of the enzyme. A crystal structure of the FL CARM1 bound to a substrate peptide is needed to test this hypothesis. We also note that functions of the N-terminal domains of the entire PRMT family remain uncharacterized[24], and in the light of the high sequence diversity of these regions, we speculate that other PRMT family members may similarly employ their N-terminal domains in substrate recognition.

Given the strong oncogenic potential of CARM1, screening for and designing CARM1 inhibitors is an active area of research, although one with the limited success[7,23]. Thus far, the structure–

activity relationship—based design of its inhibitors has focused on the catalytic domain of CARM1. However, due to the considerable degree of structural conservation among all PRMTs, SAM or sinefungin derivatives that bind this region often lack specificity. Our discovery that the N-terminus of CARM1 enables substrate recognition opens new exciting routes in the design of CARM1-specific inhibitors and warrants functional investigation of the N-terminal domains of other PRMT members, as this approach may constitute a universal path towards specific inhibitors of all PRMTs.

## Methods

**Cell lines and cell growth.** HEK293T, MCF7 and MDA-MB-231 cell lines were purchased from ATCC and routinely tested free of mycoplasma contamination. MCF7 and MDA-MB-231 have been authenticated by short tandem repeat profiling in the Translational Research Initiatives in Pathology (TRIP) laboratory at the University of Wisconsin – Madison.

HEK293T CARM1 KO, MCF7 CARM1 KO and MDA-MB-231 CARM1 KO cells were generated by transiently transfection of plasmids encoding the indicated zinc finger nuclease (ZFN) pairs; single cell was seeded into 96-well plate, and western blot and DNA sequencing were used to identify the positive clones[20]. Cells were maintained in Dulbecco's modified medium (Life Technologies, Carlsbad, CA) supplemented with 10% foetal bovine serum (Life Technologies, Carlsbad, CA). Cells were cultured at 37 °C in a humidified atmosphere containing 5% $CO_2$. For all MS experiments, cells were collected via gentle centrifugation, rinsed twice with ice-cold $1 \times$ phosphate-buffered saline (PBS; Sigma, St Louis, MO) and briefly stored at $-80$ °C before the analysis. Cells for all MS experiments were grown in biological triplicates as three individual cell cultures.

**Proteomics sample preparation and antibody enrichment.** Cell pellets were thawed on ice and resuspended in lysis buffer (8 M urea, 100 mM Tris, 10 mM TCEP, 40 mM 2-chloracetamide and protease inhibitor cocktail table; Roche, Indianapolis, IN), rigorously vortexed and sonicated in a water bath for 15–20 min. ~1,000 units of benzonase (Sigma-Aldrich, St Louis, MO) were added to fully degrade nucleic acids. Bradford protein assay (BioRad, Hercules, CA) was used to measure protein concentration. Lysate was diluted with 50 mM Tris to a final urea concentration of ~1 M before the addition of trypsin in 1:50 ratio (enzyme:protein; Promega, Madison, WI). Proteins were incubated overnight at an ambient temperature, acidified by the addition of 10% TFA, desalted over a Sep-Pak (Waters, Milford, MA) and lyophilized to dryness in a SpeedVac (Thermo Fisher, Waltham, MA). Peptides were resuspended in 0.2% formic acid, and quantitative colorimetric peptide assay kit (Pierce, Rockford, IL) was used to determine peptide concentration. Peptides were consequently lyophilized and labelled with TMT (Pierce TMT, Rockford, IL), according to the manufacturer's instruction; details on TMT channels used in each experiment are listed in Supplementary Table 5. Labelled peptides were then mixed in 1:1 ratio, and the resulting mixture was desalted over a Sep-Pak. An amount of 10–12 mg of the labelled peptide mixture was used to enrich in tandem for modified peptides using PTMScan Asymmetric Di-Methyl Arginine Motif [adme-R] antibodies (Cell Signaling Technologies, Danvers, MA), according to the manufacturer's instruction[29]. The enrichment success, that is, the number of modified peptides among all peptides detected in the precipitate, varied from 12 to 26% across all experiments. Both flow through and the eluted modified peptides were desalted and resuspended in 0.2% formic acid for LC-MS/MS analysis.

Unmodified peptides were fractionated into 20 or 32 fractions using high pH reverse phase chromatography, concatenated into 10 or 16 combined fractions, respectively, lyophilized to dryness, and resuspended in 0.2% formic acid for LC-MS/MS analysis.

**LC-MS/MS.** All capillary columns were prepared in house. A laser puller (Sutter Instruments Co., Novato, CA) was used to generate 75–360 μm inner–outer diameter bare-fused silica capillary columns with electrospray emitter tips. The tip of each column was plugged with ~5 mm of 5 μm, 130 Å pore size, bridged ethylene hybrid $C_{18}$ particles (Waters, Milford, MA). Columns were then packed with 1.7 μm diameter bridged ethylene hybrid particles to a final length of ~30 cm and installed on a Dionex Ultimate 3000 nano HPLC system (Thermo Fisher, Sunnyvale, CA), using a stainless steel ultra-high pressure union (IDEX, Oak Harbor, WA). Mobile phase buffer A consisted of water and 0.2% formic acid. Mobile phase B consisted of water, 70% acetonitrile, 0.2% formic acid and 5% DMSO. Columns were heated to 65 °C inside an in-house made heater. Peptides were loaded onto a column in 0% B and separated at a flow rate of 300–400 nl min$^{-1}$ over a 90 min gradient. Eluting peptides were analysed on a quadrupole-ion trap-Orbitrap hybrid Fusion or Fusion Lumos mass spectrometer (Thermo Scientific, San Jose, CA). Orbitrap survey scans were performed at a resolving power of 60,000 at 200 $m/z$ with an AGC target of $1 \times 10^6$ ions and maximum injection time set to 100 ms. The instrument was operated in the top speed mode with 2 s cycles and monoisotopic precursor selection turned on.

Tandem MS scans were collected in the Orbitrap at a resolving power of 60,000 at 200 $m/z$ on precursors with 2–8 charge states, using higher-energy collisional dissociation (HCD) fragmentation with normalized collision energy of 35 and dynamic exclusion of 100 s. The MS2 ion count target was set to $5 \times 10^4$ and the maximum injection time was 350 ms and 150 ms during analyses of modified and unmodified peptides, respectively.

**Data search and bioinformatics analyses.** Generated spectra were searched against the reviewed Uniprot database of human protein isoforms (downloaded 1.19.2015) and processed using the COMPASS software suite[49]. Carbamidomethylation ($+57.0513$ Da) of cysteine residues and TMT 10plex ($+229.1629$ Da) on N-termini of proteins and lysine residues were included as fixed modifications. Oxidation of methionine ($+15.999$ Da), TMT 10plex on tyrosine ($+229.1629$ Da) and dimethylation of arginine ($+28.0313$ Da) were included as variable modifications. Average mass tolerances of 125 ppm and 0.015 Da were allowed for MS1 precursor searches and MS2 fragment searches, respectively. Up to three missed cleavages on tryptic peptides with proline rule were allowed. One per cent false discovery rate (FDR) correction was performed on all identified peptides and proteins. To increase confidence in identified ADMA-containing peptides, separate 1% FDR was performed on search results that corresponded to target and decoy peptides containing the modification. TMT reporter region quantification was performed using an in-house software TagQuant, as previously described[50]. Briefly, the raw reporter ion intensity in each TMT channel was corrected for isotope impurities, as specified by the manufacturer for the used product lot, and normalized for mixing differences by equalizing the total signal in each channel. In cases where no signal was detected in a channel, the missing value was assigned with the noise level of the original spectrum (that is, noise-band capping of missing channels), and the resultant intensity was not corrected for impurities or normalized for uneven mixing.

All spectra corresponding to ADMA-containing peptides were computationally annotated using in-house software Annotated Spectrum Generator. Briefly, each peptide was fragmented in silico into its b- and y-type product ions. Using exact mass and an allowed $\pm 10$ ppm tolerance, each fragment was searched for in the associated $MS^2$ spectrum, and matched peaks were annotated with product ion type and number. All $m/z$ peaks having an intensity $<1\%$ of the $MS^2$ base peak were eliminated from consideration. All charge states less than the parent precursor charge state were considered when matching fragments to $m/z$ peaks in the $MS^2$ spectrum. Ions produced on the characteristic neutral loss of dimethylamine ($-45.0837$ Da) were identified by subtracting the lost mass off the mass of arginine residues carrying ADMA modification and allowing for charge-state changes.

$P$ values, associated with all fold-change measurements, were calculated using two-tail Student's $t$-test. To correct for multiple hypothesis testing, generated $P$ values were converted into $q$ values using Storey method[51]. Q value of 0.01 corresponded to FDR of 1%. Hierarchical clustering was performed using Pearson correlation and average linkages after preprocessing with 10–15 k mean clusters. Clustering and data visualization were carried out in Perseus[52]. Reactome database search[39] and combined score calculations were performed using Enrichr online software[53]. Briefly, the combined score was calculated by multiplying log transformed $P$ value computed using the Fisher exact test with the z-score of the deviation from the expected rank; higher score indicates stronger enrichment. STRING database[40] was searched for interactions among identified substrates (v 10.0, accessed 9 December 2016). Data from COSMIC[38] were accessed and downloaded from the website (v 77, accessed 8 October 2016). Analysis of amino-acid abundances was conducted by comparing previously published data[54] on all human proteins and percentages calculated over an 11 amino-acid segment centred on the dimethylated arginine residue detected in this study. MotifX online software was used to extract enriched motifs[43]. Motif width of 13 residues and its minimum representation in ~15% of detected sites at the significance of $P$ value of less than $10^{-5}$ were required to detect an over-represented motif.

**Purification of Halo-tagged proteins.** The FL and N-terminal domain-deleted version of human CARM1 cDNAs, and other PRMT FL cDNAs were cloned into pFN21K Halo Tag CMV Flexi mammalian expression vector (Promega, Madison, WI) using the unique SgfI and PmeI site[55]. To express the protein, DNA constructs were transiently transfected into HEK293T CARM1 KO cells in 15 cm dishes using transIT-LT1 reagent (Mirus Bio, Madison, WI) according to the manufacturer's protocol. Proteins were allowed to express for 48 h. Cell pellets were then collected and lysed in lysis buffer (50 mM HEPES, pH 7.5, 150 mM NaCl, 0.005% NP-40, 0.5 mM EDTA, supplemented with $1 \times$ protease inhibitor cocktail before use). The cells were sonicated on ice using a Branson Sonifier 450 with a microtip at 35% amplitude (10 s on, 30 s off, 6 cycles). The crude lysate was cleared by centrifugation (15,000 r.p.m., 4 °C, 20 min). The cleared lysate was then mixed with pre-washed Halo Link resin (60 μl resin for a 15 cm dish) and incubated with rotation at room temperature for 2 h. The resin was then washed twice with lysis buffer, once with lysis buffer supplemented with 1 M urea and twice with lysis buffer. The recombinant proteins were then eluted with cleavage buffer (lysis buffer

supplemented with 1 mM DTT and 20 µg ml$^{-1}$ TEV protease) with rotation at room temperature for 2 h. The resin was centrifuged at 3,000 r.p.m. for 5 min, and the supernatant containing the cleaved protein was collected and protein concentration was determined using the Bradford assay (Thermo Scientific, Rockford, IL). Protein purity was assessed by running SDS–PAGE and Coomassie Blue staining.

**Purification of Flag-tagged proteins.** CMX-Flag-BAF155 construct was generated by subcloning a PCR-amplified Flag-BAF155 fragment into the CMX-expression vector. DNA construct was transiently transfected into HEK293T CARM1 KO cells in 15 cm dishes using transIT-LT1 reagent (Mirus Bio, Madison, WI) according to the manufacturer's protocol. The protein was allowed to express for 48 h. Cell pellets were then collected and lysed in lysis buffer (50 mM Tris, pH 8.0, 150 mM NaCl, 10% glycerol, 0.5% Triton X-100, supplemented with 1 × protease inhibitor cocktail before use) with 30 min rotation at 4 °C. The crude lysate was cleared by centrifugation (15,000 r.p.m., 4 °C, 20 min). The cleared lysate was then mixed with pre-washed ANTI-FLAG M2 Affinity Gel (60 µl resin for a 15 cm dish) and incubated with rotation at room temperature for 2 h. The resin was then washed 3–5 times with lysis buffer and then the recombinant protein was eluted with elution buffer (lysis buffer supplemented with 150 ng µl$^{-1}$ 3xFLAG peptide) with rotation at room temperature for 1 h. Protein concentration was determined using Bradford assay and protein purity was assessed by SDS–PAGE and Coomassie Blue staining.

**Purification of GST-tagged proteins from bacteria.** The DNA fragments were cloned into pGEX-2T GST expression vector (GE Healthcare, Waukesha, WI). DNA constructs were transformed into BL21(DE3)pLysS E. coli expression strain. Luria–Bertani media (1–2L) was seeded with 10 ml of overnight starter culture and grown at 37 °C until OD$_{600\,nm}$ reached 0.6. Protein expression was then induced with the addition of 0.2 mM isopropyl β-D-1-thiogalactopranoside for 4 h at ambient temperature. Bacterial pellets were then lysed by sonication on ice in lysis buffer (50 mM Tris, pH 7.5, 150 mM NaCl, 0.05% NP-40, supplemented with 1 × protease inhibitor cocktail before use) with 70% amplitude, 30 s on, 30 s off, 10 cycles. The crude lysate was cleared by centrifugation at 13,000 r.p.m., 4 °C for 30 min. The cleared lysate was then mixed with pre-washed glutathione resin (Thermo Scientific, Rockford, IL) and incubated with rotation at ambient temperature for 2 h. The resin was then washed three times with lysis buffer and once with 100 mM Tris (pH 8.0). The recombinant protein was then eluted with elution buffer (100 mM Tris, pH 8.0, 15 mg ml$^{-1}$ glutathione) with rotation at ambient temperature for 1 h. Eluted protein was dialysed in snake skin dialysis tubing (Thermo Scientific, Rockford, IL) in dialysis buffer (50 mM HEPES, pH 7.5, 150 mM NaCl, 0.005% NP-40, 0.5 mM EDTA) at 4 °C for overnight. The following day, purified protein was retrieved from the dialysis tubing and concentrated to desired concentration. Protein concentration was then determined using Bradford assay and protein purity was assessed by SDS–PAGE and Coomassie Blue staining.

**Purification of 6xHis-tagged proteins from bacteria.** The DNA fragment encoding CARM1 residues 28–140 was cloned into pET21a expression vector (Novagen, Madison, WI). DNA construct was transformed into BL21(DE3)pLysS E. coli expression strain. Luria–Bertani media (1–2L) was seeded with 10 ml of overnight starter culture and grown at 37 °C until OD$_{600\,nm}$ reached 0.6. Protein expression was then induced with the addition of 0.2 mM isopropyl β-D-1-thio-galactopranoside for 4 h at ambient temperature. Bacterial pellets were then lysed by sonication on ice in lysis buffer (50 mM Tris, pH 7.5, 150 mM NaCl, 0.05% NP-40, supplemented with 1 × protease inhibitor cocktail before use) with 70% amplitude, 30 s on, 30 s off, 10 cycles. The crude lysate was cleared by centrifugation at 13,000 r.p.m., 4 °C for 30 min. The cleared lysate was then mixed with pre-washed TALON metal affinity resins (Clontech, Mountain View, CA) and incubated with rotation at 4 °C for overnight. The resin was then washed once with lysis buffer and three times with lysis buffer containing 20 mM imidazole. The recombinant protein was then eluted for three times with elution buffer (lysis buffer containing 300 mM imidazole) with rotation at 4 °C for 10 min. Eluted protein was dialysed in 3.5 kDa snake skin dialysis tubing (Thermo Scientific, Rockford, IL) in dialysis buffer (50 mM HEPES, pH 7.5, 150 mM NaCl, 0.005% NP-40, 0.5 mM EDTA) at 4 °C for overnight. The following day, purified protein was retrieved from the dialysis tubing and concentrated to desired concentration using 3 kDa MWCO 50-ml centrifugal concentrator. Protein concentration was then determined using Bradford assay and protein purity was assessed by SDS–PAGE and Coomassie Blue staining.

**Western blotting.** Cells were collected by trypsinization, washed with Dulbecco's phosphate buffer saline (Life Technologies, Carlsbad, CA) and lysed by suspension in lysis buffer (50 mM Tris-HCl pH 8.0, 150 mM NaCl, 10% glycerol, 0.5% Triton X-100 and 1 × protease inhibitor cocktail (Sigma-Aldrich)). After brief sonication followed by centrifugation, total protein was quantified using the BioRad Protein Assay (BioRad, Hercules, CA), and 20 µg protein was resolved by SDS–PAGE. Proteins were transferred to a nitrocellulose membrane for 1.5 h at 350 mA. Membranes were blocked with 5% nonfat milk at room temperature for 1 h and incubated overnight with diluted primary antibody at 4 °C. Membranes were then

washed and incubated with HRP-conjugated secondary antibody for 1 h at room temperature. Signal was visualized by enhanced chemiluminescence reagents (Thermo Scientific, Rockford, IL) and X-ray film exposure followed by scanning and digitalization. Me-PABP1 and PABP1 antibodies[55], me-BAF155 antibody[20] and me-MED12 antibody[30] were generated previously in the Xu lab. FLAG antibody (catalogue #F7425) was purchased from Sigma and used at a dilution of 1:5,000; BAF155 antibody (sc-10756) was purchased from Santa Cruz and used at a dilution of 1:1,000; MED12 antibody (ab70842) was purchased from Abcam (Cambridge, MA) and used at 1:1,000 dilution.

**FP assay.** Fluorescein-labelled BAF155 peptide (constant final concentration at 5 nM) was mixed with titrations of purified recombinant proteins or BSA control in binding buffer (50 mM HEPES, pH 7.5, 150 mM NaCl, 0.005% NP-40, 0.5 mM EDTA) and incubated in dark at room temperature for 2 h. A no-protein control with only peptide probe and a blank buffer-only control were included. Polarization at each concentration was measured as triplicates in 384-well polystyrene black microplates (Thermo Fisher Scientific #262260) by Biotek Synergy H4 multimode plate reader (light source: xenon flash, offset from top: 7 mm, sensitivity: 60%, excitation: 485/20 nm, emission: 528/20 nm, both parallel and perpendicular, normal read speed). Polarization measurements were then corrected for background contributions to the measured intensity by subtracting the parallel and perpendicular intensity readings, from the blank buffer-only wells, from the intensity readings for each data point. Quantitatively, FP was calculated as the difference of the emission light intensity parallel (I||) and perpendicular (I⊥) to the excitation light plane normalized by the total fluorescence emission intensity by equation: $FP = 1,000 \times (I|| - I\perp)/(I|| + I\perp)$, given in units of millipolarization.

**In vitro methylation assays.** Purified recombinant enzyme and substrates were incubated in methylation buffer (50 mM Tris-HCl, pH 8.0, 100 mM NaCl and 0.5 mM EDTA) for 1–4 h at 30 °C in the presence of 1 µl radioactive S-adenosyl-L-[methyl-$^3$H] methionine. The reaction was stopped by adding 5 × SDS sample buffer and boiling at 99 °C for 5 min. Proteins were resolved by SDS–PAGE. The gel was stained with Coomassie Brilliant Blue (Coomassie Brilliant Blue R-250 0.05% w/v: methanol 50%: acetic acid 10%: H$_2$O 40%) followed by de-staining with dye-free buffer. Fixed gel was then soaked with amplify (GE Healthcare, Waukesha, WI) with gentle shaking at room temperature for 30 min before drying on a Whatman paper at 80 °C and exposing to an X-ray film in dark at −80 °C. Protein methylation was detected by autoradiography.

For methylation assays using the peptide array, 16 or 20 µg purified recombinant CARM1 or same molar concentration of the other PRMTs (for 96-spot and 192-spot arrays, respectively) was mixed well with 40 µl radioactive S-adenosyl-L-[methyl-$^3$H] methionine, 80 µl 5 × methylation buffer and ddH$_2$O to achieve a final volume of 200 or 400 µl reaction (for 96-spot and 192-spot arrays, respectively). The Celluspot peptide arrays were customized and produced by Intavis Bioanalytical Instruments, Germany. Before the methylation assay, the peptide area on the array slide was circled using a hydrophobic PAP pen. The mixed methylation reaction was then applied onto the array slide to cover the circled peptide area, and then the slide was placed in a humidified chamber and incubated at 30 °C in the hybridization oven for 3 h. The methylation reaction was then gently flicked off and the slide was rinsed with 1 × methylation buffer. The array slide was washed in methylation buffer for three times and then soaked in amplify solution for 30 min with rotation at room temperature. The slide was then washed again with methylation buffer for three times and air dried before exposure to X-ray film at −80 °C for 3 days. A well that contained no peptide sequence was used as the negative control and the quantification reference.

**Virus packaging and stable cell line generation.** Virus packaging was performed by transfection of three plasmids (PHIT60, pLNCX-PRMT expressing plasmid, VSVG) into HEK293T cells. Supernatant containing the virus was collected for cell line infection after 48 h (ref. 20). To generate the stable cell lines, $2 \times 10^5$ MCF7 CARM1 KO cells were seeded into six-well plate 24 h before infection. For infection, 1 ml virus was mixed with 1 ml fresh cell culture medium; polybrene was added at a final concentration of 5 µg ml$^{-1}$ to increase the infection efficiency. Cells were infected overnight before cell culture medium was changed. Cells were selected with 400 µg ml$^{-1}$ G418 for at least 4 weeks to obtain the stable cell lines.

**Data availability.** All raw mass spectrometric data files and annotated spectra supporting the findings of this study are available through Chorus (ID 1174) and the ProteomeXchange Consortium via the PRIDE partner repository[34] (accession #PXD005871). Original, uncropped images of blots and gels are provided as Supplementary Fig. 6. All other data are available within the paper or included as its Supplementary Materials, and available from the corresponding author on request.

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

## Acknowledgements

We thank Dr Jonathan Florentin for assistance with peptide array design and Dr Lu Wang for initial involvement in the project. We gratefully acknowledge support from the National Institutes of Health Grant P41 GM108538 (awarded to J.J.C.) and NIH R21CA196653 and NIH R01 CA213293 (awarded to W.X.). This work was also supported in part by NIH/NCI P30CA014520-UW Cancer Center Support Grant and U54DK104310.

## Author contributions

E.S. designed and conducted mass spectrometry experiments, interpreted results and performed bioinformatics analyses. H.Z. performed and interpreted biochemical experiments, in vitro methylation assays and generated mutant cell lines. F.L. assisted with construct design, stable cell line generation and performed in vitro methylation assays. N.W.K. developed a computation approach to spectrum annotation and assisted with bioinformatics analyses. A.S.H. assisted with design, optimization and interpretation of mass spectrometry experiments. J.J.C. and W.X. directed and supervised all aspects of the study. All authors contributed to writing and critical review of the manuscript.

## Additional information

**Competing interests:** The authors declare no competing financial interests.

