## [Peer Review File · Nature Communications]

Reviewers' Comments:

Reviewer #1 (Remarks to the Author):

Shishkova et al identify over 300 unique ADMA methylated sites of which 130 represent new CARM1 substrates with nano-liquid chromatography tandem mass spectrometry (nanoLC-MS/MS). These CARM1 substrates confirm the previously known proline-like motif specificity of CARM1 and map the N-terminus of CARM1 as a necessary and sufficient motif for their recognition. They propose to target the N-terminus to develop CARM1 inhibitors.

The identification of the 130 CARM1 substrates using 2 breast cancer cell lines is important. However, this manuscript is descriptive and does not provide much new mechanistic insight into the function of substrates. The mapping of the N-terminal domain as a regulatory sequence of CARM1 is a significant finding, but also does not prove that such inhibitors can and will be useful.

Reviewer #2 (Remarks to the Author):

In the manuscript by Shishkova et al, the authors perform a quantitative proteomics experiment using TMT-multiplexing in order to identify novel substrates of the arginine methyltransferase CARM1 (also known as PRMT4). To this end, the authors use wild-type cells and quantitatively compare these to CARM1 KO cells for the identification of regulated arginine methylation sites, which should reside on arginine residues targeted by CARM1 in wild-type cells only. Following this, the authors report a proline-rich motif targeted by CARM1, and that CARM1 requires its N-terminal tail for substrate recognition.

Although the manuscript is well written, and the experimental design is elegant, the data presented by the authors contains several critical aspects that needs to be addressed in order for the manuscript to be considered for publication in Nature Communications.

Generally, the mass spectrometric data needs to be further validated from a quantitative aspect in order to determine whether the experimental setup indeed is able to identify regulated CARM1 substrates. Secondly, the bioinformatic analyses are unintentionally biased and need to be redone in a properly manner, while the in vitro confirmation of identified CARM1 substrates contains several critical points that raise concern about its validity.

Major concerns:

- For identification of regulated ADMA sites the authors use TMT6-plex labeling WT and KO cells in triplicates. However, the authors fail to convince that their identified ADMA sites indeed are quantitatively regulated between investigated cells. They show a nice heat map (Figure 1B) of identified ADMA sites, and that certain regulation is observed using label-free quantification (Figure 1C). However, the open question is how does all the unmodified peptides compare to this? Are the unmodified peptides identified throughout this experiment quantitatively scattered in a manner similar to the ADMA containing peptides? In order to address this, the authors need to include three analyses:

1) First the authors have to include scatter plots between their individual experiments to demonstrate that a strong Pearson correlation is observed between replicate experiments. The authors have nicely done this in the last part of the manuscript, when comparing the N-terminal truncated CARM1 to KO cells (Figure 4F). A similar comparison needs to be included for all identified peptides in each experiment (with all peptides referring to both ADMA modified and unmodified peptides).

2) The authors need to prepare a Volcano plot for all identified peptides (ADMA modified and unmodified peptides), analogous to the data presented in Figure 1C. In this figure, the authors can then specifically highlight the ADMA containing peptides, which should be significantly regulated as compared to unmodified counterparts. Otherwise the quantitative aspects of the experiment cannot be used to infer which proteins are CARM1-substrates in an unbiased manner.

3) Please prepare a heat map analysis for all unmodified peptides as well in a similar manner to the heatmaps presented in Figure 1B, 4E and S1B.

- From the experimental setup the authors used three replicates to obtain their results. However, the manuscript does not state whether these replicates are biological or technical replicates? Do the analyzed cells stem from three independent experiments, or do they stem from the same experiments being split into three?

- The authors only use FDR correction at the peptide level for ADMA peptides. FDR on all identified peptides should be included!

- How clean are the ADMA enriched samples? How many percent of the total number of identified peptides in the experiments are actually identified as ADMA peptides? Please include such evaluation data.

- The authors look into changes in overall protein abundance of the identified substrates. However, in this analysis a decrease in protein abundance is observed for several of the other PRMTs (especially since PRMT1 and 6 are both type I enzymes that catalyze ADMA). Please

explain whether a decrease in ADMA levels could be connected to a decrease in expression of these type I PRMTs?

- Besides, several of the known PRMTs are not identified in the MS analysis – please include a WB for all different PRMTs (at least for all type I PRMTs that preferentially catalyze ADMA) for WT and KO CARM1 cells to determine the expression levels of these in the experiment setup.

- For the bioinformatic analyses presented in Figure 1D, Figure 2A and 2B the authors seem to be making an unintentional mistake in their comparison between ADMA containing arginines and unmodified arginine residues. All the presented data seems to stem from comparisons between identified ADMA sites and essentially remaining arginine residues in the human proteome. However, such an approach may unfortunately lead to false biases as the proteomics experiment certainly is abundance biased – i.e. any proteomics experiment has a strong tendency for identifying more abundant proteins as compared to low abundant ones. Consequently, the observed difference between ADMA sites and arginine residues (AnyR) may simply reflect a difference in the proteins being compared and not necessarily a difference related to ADMA. For example, the identified ADMA sites reside on proteins involved in RNA-associated processes, and these proteins have been described to harbor a larger number of somatic mutations as compared to other proteins. Hence, the observed difference showed in Figure 1D may simply be due to the ADMA sites primarily being identified on RNA-associated proteins – which their analysis strongly support.

To evaluate this, and remove any possible abundance or protein-group bias, the authors will have to redo their analyses and only compare arginine residues residing on the same proteins. That is, for a given protein X where an ADMA site has been identified the authors need to compare this ADMA site to a randomly chosen arginine in protein x that is not methylated. This is the only way how any abundance bias can be removed. Moreover, this way the authors also ensure that the two datasets being compared (ADMA sites versus nonmodified sites) actually contain that same number of sites.

- Likewise, the analyses shown in Figure 2A and 2B needs to be done in a similar way as described above. Comparison of one ADMA site on protein x to a randomly chosen unmodified arginine on the same protein x. Especially considering that the observations that the authors want to report may in fact just stem from the abovementioned abundance bias.

- For the in vitro experiments described in Figure 2C, 2D and 2E, the authors need to include proper controls. For example, not positive or negative control is included in the analyses for PRMT1, PRMT5 or PRMT6 shown in figure 2E. Hence, the observed difference between figure 2E and 2D may just simply stem for the various blots being exposed differently. The authors

therefore need to include such controls analogous to those included in Figure 2C.

- Likewise, the authors will have to comment on why there are so many empty spots for the identified CARM1 substrates shown in Figure 2D? Identification of substrates via their MS experiment should entail a 1% FDR, but from figure 2D it seems that a large number of spots show no real coloring (i.e. does not constitute proper CARM1 substrates). And many of these empty spots even seem reproducible between the two replicates?

Minor points:

- Please include a rationale why the MS experiments were not performed using MS3 to avoid ratio compression? Especially considering that the data was measured on an Orbitrap Lumos Fusion instrument which is fully capable of analyzing MS3?

- Please include resolution used for MS/MS scans.

- The collision energy used seems rather high. Any reasons for this?

- In the methods section the authors describe that they used TMT10-plex, but the data described in figure one is only TMT6-plex?

- On page 2 the authors claim that mono-methylation is only catalyzed by PRMT7. However, it is widely known that the majority of PRMT enzymes can catalyze mono-methylation (Bedford and Clarke, 2009), including CARM1. However, certain PRMT enzymes primarily catalyze di-methylation, but has still been reported to catalyze mono as well.

- The authors use both MCF7 and the triple negative cell line MDA-MB-231. They find less CARM1 substrates in MDA-MB-231 cells, but they do not at any point comment on possible explanations for this finding. (And they could maybe state that only the MCF7 cells are used for the rest of the experiments and why this is).

- Figure 3C; please include a loading control.

- Figure 4E; there seems to be an increase in abundance of some ADMA peptides for both CARM1 140-608 cells and CARM1 KO cells (maybe even more in the CARM1 140-608 cells compared to the KO cells). The authors mention in the main text that the observed increase in abundance could be due to a decrease in PRMT1. However, at the same time they state that no decrease in PRMT1 abundance is observed in CARM1 140-608 cells. Please explain? Could the observed difference simply constitute an artifact of using z-scoring for the generation of heat-maps?

- Figure 1F: How does the Pearson correlation look like when comparing CARM1 140-608 to wild-type? This correlation should be similar to the one that the authors need to include for each of their initial WT vs KO experiments (as mentioned in the first point of these comments).

Reviewer #3 (Remarks to the Author):

In the current manuscript the authors use a quantitative MS-based proteomics approach to carry out an unbiased de novo and peptide-specific identification of CARM1 substrates in breast cancer cells. This led to the identification of 130 putative CARM1 protein substrates, the vast majority of which was validated in vitro through an ad hoc built peptide-array whereby a list of 200 peptides (among experimentally identified and predicted ones) was tested as substrates for enzymatic methylation assays using recombinant CARM1 and other PRMTs as control to confirm specificity. Additionally, bioinformatics analysis of the newly identified sites allow to definition of a putative novel CARM1 recognition motive, whereby the enrichment of Proline in the vicinity of methylated Arginine is confirmed whereas Glycine and Methionine not. In the second part of the paper the authors focus on analyzing the functional role of the N-terminal domain of CARM1 and with a set of biochemical in vitro assays demonstrate that it is necessary for the efficient recognition, binding and methylation of the Proline/Arginine -rich motifs. Hence, this study extends the knowledge about the activity and substrate recognition of this important enzyme (which is aberrantly expressed in many cancers and is a promising target for therapy through the development of inhibitors), thus providing a useful resource for the design of novel, more specific inhibitors that may take into account the newly discovered N-terminal domain function.

The topic addressed by this study is undoubtedly relevant, interesting and very hot and the MS-proteomics- approach may in principle extend significantly the knowledge on the molecular activity/specificity of CARM1 protein-methyl-transferase, an enzyme aberrantly expressed in many cancer types. It is overall well designed and well written, although the two parts composing the paper (the MS-proteomics dissection of the CARM1- substrates and the biochemical analysis of the function of CARM1 N-terminal domain) appear to be quite independent/disjointed/separated.

In spite of its scientific relevance and novelty and the overall innovative approach, the paper suffers from some major (and minor) limitation that must be addressed to improve quality, reliability of the data and the overall impact of the study.. Such issues are outlined in the points below.

Major issues:

- 1) The major novelty of the paper lays on the of unbiased identification of about 300 novel CARM1 -target sites on almost 140 proteins through a MS-based proteomics approach based on

sample fractionation, asymmetric di-methylated peptide enrichment by immunoaffinity (anti-pan-ADAM antibody from CST) and TMT-based quantitative analysis of methyl-variations upon CARM1 KO. The approach followed is based very recent technical advancements in the field of the methyl-proteome analysis by MS. However, precise details about the MS-data and their analysis, which led to the identification of the 300 sites, are totally absent. I am aware that Nat Comm. is NOT a proteomics journal; nevertheless, because the major novelty of the manuscript resides in the collection of these new sites, it is mandatory to provide (as supplemental or through a repository) all MS/MS fragmentation spectra for the newly identified methylated peptides, to allow scientists judging the reliability of the information extrapolated. This is especially relevant since the authors state that they have carried out the validation of the novel asymmetric-di-methylated sites peptide through the manual annotation of the fragment peptides and the manual identification of the marker ion derived from the neutral loss of DMA. Provide the MS/MS fragmentation spectra for at least for a reasonable subset (at least 30%) of these peptide peptides is therefore essential. Fragment ions and peptide coverage should be clearly displayed to demonstrate both that the methylation is precisely assigned and that the specific report ion for ADMA is always detectable, as stated.

Indeed the neutral loss ion fragment, which the authors refer to, is described in C. J. Brame et al , where the authors state that: “We speculate these ions arise from neutral loss of monomethylamine, dimethylcarbodiimide, and dimethylamine”... Actually, these are not very well-established diagnostic ions for MMA/ADMA by the community and have not been extensively employed by other groups working on methyl-proteomes to discern between ADMA/SDMA. Hence, the authors should provide not only the manually annotated MS/MS spectra containing these diagnostic ions, but also many more details on the analysis of MS data in Experimental section.

2) The second major limitation concerns to the experimental evidence of Suppl. Figure S1C, that PRMT1 is down-regulated 2.5 fold by CARM1 KO in MCF7 cells. This is a very major problem that could jeopardize the analysis, given that PRMT-1 is the major type -I PRMT, accountable for more than 80% of the R-methyl-proteome. A 2.5 fold decrease of this enzyme is a very significant change and must have a remarkable impact on the set of the asymmetric-di-methylated peptides investigated in this study. By no means imposing a 2-fold change cut-off in the methyl-peptide abundance can provide a solution for this specificity problem as proposed by the authors. This is a quantitative filtering criterion that does not distinguish and discriminate for substrate specificity! Since PRMT1 down-regulation appears to be cell-type dependent -and in fact it does not occur in MDA-MB-231 cells- the authors should r-select breast cancer cell lines that do not display PRMT1 down regulation upon CARM1 KO, like the MDA-MB-213 cell line and exclude MCF7 from the proteomic screening. On this line, remarkably the authors do not specify whether their list of 300 methylated peptides is derived from the INTERSECTION or the UNION of the two screenings made on MCF7 and MDA-MB-231, respectively: while the

intersection (all peptide are methylated and down regulated >2 fold upon KO in BOTH cell lines) would be acceptable, the UNION is not good and the peptide that are downregulated in MCF7 only are potentially false positive and should be excluded.

3) Strictly related to this is the remark made by the authors in the last paragraph of the results, when they comment on the up regulation of a minor subset of asymmetric di-methylated peptides upon CARM1 KO and CARM1 N-terminal deletion mutant and propose the explanation that PRMT1 is also down regulated and hence substrate scavenging activity by other PRMTs could take place on the R-sites set free from depleted PRMT1 (as described in Dhar. et al.). This hypothesis challenges the whole proteomics screening in Figure 1, because it implies that PRMT1 downregulation does have a relevant effect of the methyl-proteome and that such effect may be composite and not easy to be dissected. In fact it can lead to either non compensated down-regulation of ADMA-peptides (which can be misinterpreted as false positive CARM1 substrates –as discussed), or effect which will be compensated and thus non-changing and non detectable, leading to false negatives. In any case, this highlights the limit of the current study and corroborates the need to carry out these proteomics –based screening for CARM1 peptides in model systems where not other PRMTs are affected.

4) With this high risk of false positive in the putative list of novel CARM1 target selected through the -2fold change cutoff, the in vitro validation experiment using the selected peptide array (Figure 2C-E) becomes vital to validate the MS data. Although this could be in principle a well designed assay, it misses important controls namely positive controls peptides for PRMT1, PRMT5 and PRMT6 activity, to be included within the same array in order to clearly demonstrate that the enzymatic activity of these other PRMTs is comparable to that of CARM1. The results displayed in figure 2E are not convincing since it is not possible to: a) confirm the activity of the enzymes b) compare it directly with that of CARM1 within the same array, on a common set of substrates. This experiment, must then be repeated, in a modified form, to include these controls.

5) In the second part of the manuscript (figures 3 and 4 and associated supplementary data), all in vitro methylation experiments are carried out with BAF155, MED12, PABP1 as substrates. These are already known CARM1 substrates, well characterized in previous publications by the authors. It would be nice, (also with the aim of better linking the two parts of the story), to include in the same experiments a few of the substrates newly identified through the MS-screening. Additionally, it is unclear why in this part HEK293T wt and KO cells were used, given that the initial screening of CARM1 target had been carried out in breast cancer cell lines. This contributes to make the two parts of the manuscript even more disconnected/disjointed.

Minor Points

1) Since HpH based-fractionation before immuno-enrichment of methylated peptides has been

shown to highly increase the detection of methyl-peptides (Ref 2), it is unclear why the authors chose to fractionate just the unmodified peptides fraction through this method. Indeed, it would be much more worth fractionating the cellular peptidome before the methyl-peptides immunoenrichment with pan-ADMA-peptides, as described in recent publications.

2) The figure 1B shows the unnormalized log₂ TMT ratios of the identified methyl-peptides and the corresponding protein ratio are displayed in a separate figure (S1C). However, it is advisable to normalize the TMT ratios of each modified peptides to the respective protein level and then display and analyze this normalized/corrected ratio, which would take into account for each modified peptide even any minor contribution of the respective protein abundance.

3) In Figure 1B and S1B, the authors show that 50% of the identified methyl-peptides are regulated in CARM1 KO MCF7 cells and MDA-MB-231, respectively. However, in suppl. Table S1 several methyl-peptides contain more than one “methylatable” site, making the assignment of the observed regulation at the single site level difficult. Although we are aware that the site-specific assignment is challenging, this is an important point that may affect the motif-analysis and should be addressed and discussed by the authors.

4) In general, figures containing the MS workflow, the MS spectra and the clustering analysis of TMT-labelled methyl peptides are too small and not really readable.

5) The MS analysis is overall good quality, in spite of the limitation outlined at point 1. However, as said, the description of the Experimental procedures must be extended, with more information on the TMT quantification (e.g. the TMT labels used, the number of replicates, the experiment setup), the report ion analysis and detection from the neutral loss of ADMA.

6) Wrong numbering of Figures S4 E and S4 E in the main text.

7) In the first paragraph of the results, the sentence “..Our experimental design capitalized on the multiplexing capabilities of tandem mass tags (TMT), a technique that allowed us to enrich and analyze ADMA-modified peptides from several samples simultaneously..” is not formally correct as it delivers the concept that the use of TMT-labeling serves for methyl-peptide enrichment and detection. The TMT indeed is only useful for the quantitative labeling, and in particular for its multiplexing capability, which the authors did not take advantage of. This sentence should be therefore corrected.

8) Overall is quite amazing – to be honest and a bit surprising – that the quantitative comparison of the asymmetric-dimethyl-peptides changes upon CARM1 KO and CARM1 N-terminal (140-608) mutant shows that the mutant seem to have almost a “linear” effect on methylation, exactly intermediate between wt and KO (figures 4E and F). This kind of linear/dose-dependent effects -

affecting similarly most R-sites profiled, on most of the substrates analyzed- is rather unusual in global-MS-modification studies, so it somehow hard to imagine. Although we obviously do not question the accuracy of TMT quantification, maybe the authors could argue more about this impressive linearity of response.

In light of all this considerations, we think that the manuscript has potential for publications in this journal, but only upon major revision that will address the points outlined.

We thank reviewers and the editor for their valuable input and the opportunity to resubmit this manuscript upon major revisions. We believe that reviewers' critical feedback was instrumental in improving the manuscript to the point where the quality of the final product far exceeds that of the original submission. We conducted additional experiments that tested reviewers' specific suggestions, modified main figures, and introduced new supplementary figures. The current draft also incorporates all the relevant experimental details that may have been omitted from or presented unclearly in the original draft of the manuscript. Please see the response below for more details. The reviewers' comments are in black, and our responses are in blue.

Reviewer #1 (Remarks to the Author):

Shishkova et al identify over 300 unique ADMA methylated sites of which 130 represent new CARM1 substrates with nano-liquid chromatography tandem mass spectrometry (nanoLC-MS/MS). These CARM1 substrates confirm the previously known proline-like motif specificity of CARM1 and map the N-terminus of CARM1 as a necessary and sufficient motif for their recognition. They propose to target the N-terminus to develop CARM1 inhibitors.

The identification of the 130 CARM1 substrates using 2 breast cancer cell lines is important. However, this manuscript is descriptive and does not provide much new mechanistic insight into the function of substrates. The mapping of the N-terminal domain as a regulatory sequence of CARM1 is a significant finding, but also does not prove that such inhibitors can and will be useful.

We are happy that the reviewer found our findings important and significant, and we hope that future studies by our laboratories and other research groups will alleviate the concern over the utility of proposed inhibitors.

We agree that our manuscript does not provide functional insight into the mechanism of novel CARM1 substrates; however, this was not the main intent of the work. Prior to our study, very few CARM1 substrates were known. We have provided a nearly 10-fold boost in known substrates. This achievement by itself is quite significant. We also leveraged these results to provide some novel insight into how CARM1 functions and recognizes substrates. These two contributions are considerable, may help in development of CARM1 inhibitors, and in our view justify publication in this high profile venue.

Reviewer #2 (Remarks to the Author):

In the manuscript by Shishkova et al, the authors perform a quantitative proteomics experiment using TMT-multiplexing in order to identify novel substrates of the arginine methyltransferase CARM1 (also known as PRMT4). To this end, the authors use wild-type cells and quantitatively compare these to CARM1 KO cells for the identification of regulated arginine methylation sites, which should reside on arginine residues targeted

by CARM1 in wild-type cells only. Following this, the authors report a proline-rich motif targeted by CARM1, and that CARM1 requires its N-terminal tail for substrate recognition. Although the manuscript is well written, and the experimental design is elegant, the data presented by the authors contains several critical aspects that needs to be addressed in order for the manuscript to be considered for publication in Nature Communications.

Generally, the mass spectrometric data needs to be further validated from a quantitative aspect in order to determine whether the experimental setup indeed is able to identify regulated CARM1 substrates. Secondly, the bioinformatic analyses are unintentionally biased and need to be redone in a properly manner, while the in vitro confirmation of identified CARM1 substrates contains several critical points that raise concern about its validity.

We thank the reviewer for the strong support of our work and findings. Below we address the critical aspects raised and are confident that the manuscript is now greatly improved.

Major concerns:

- For identification of regulated ADMA sites the authors uses TMT6-plex labeling WT and KO cells in triplicates. However, the authors fail to convince that their identified ADMA sites indeed are quantitatively regulated between investigated cells. They show a nice heat map (Figure 1B) of identified ADMA sites, and that certain regulation is observed using label-free quantification (Figure 1C). However, the open question is how does all the unmodified peptides compare to this? Are the unmodified peptides identified throughout this experiment quantitatively scattered in manner similar to the ADMA containing peptides? In order to address this, the authors need to include three analyses:

1) First the authors have to include scatter plots between their individual experiments to demonstrate that a strong Pearson correlation is observed between replicate experiments. The authors have nicely done this in the last part of the manuscript, when comparing the N-terminal truncated CARM1 to KO cells (Figure 4F). A similar comparison need to be included for all identified peptides in each experiment (with all peptides referring to both ADMA modified and unmodified peptides).

Below are the plots demonstrating reproducibility between biological replicas of wild type and knockout samples of MCF7 cells. Abundance measurements of both ADMA-containing (in red) and unmodified (in black) peptides correlated very well across the three replicas of each sample with median R^2 of 0.98. Similar or better reproducibility was achieved in experiments using MDA-MB-231 cells and MCF7 cells expressing N-terminus truncated CARM1 (please see Supplementary Table S1 below) with median R^2 of 0.98 across all sets of biological replicas in all three TMT experiments.

As evident from the plots, both modified and unmodified peptides were measured with equally good reproducibility. Such high data quality ensured that calculated fold changes were accompanied with confident measurements of statistical significance, and we employed both of these parameters to distinguish putative CARM1 substrates from substrates of other PRMTs.

The scatter plots were added as NEW Supplementary Figure S1A.

MDA-MB-231 wild type (WT)

WT1	NA		
WT2	.98	NA	
WT3	.98	.99	NA
	WT1	WT2	WT3

MDA-MB-231 knockout (KO)

KO1	NA		
KO2	.98	NA	
KO3	.98	.99	NA
	KO1	KO2	KO3

MCF7 CARM1 28-608 (WT)

WT1	NA		
WT2	.99	NA	
WT3	.98	.97	NA
	WT1	WT2	WT3

MCF7 CARM1 28-608 (WT)

WT1	NA		
WT2	.98	NA	
WT3	.99	.99	NA
	WT1	WT2	WT3

MCF7 CARM1 140-608 (TR)

TR1	NA		
TR2	.99	NA	
TR3	.98	.98	NA
	TR1	TR2	TR3

MCF7 knockout (KO)

KO1	NA		
KO2	.98	NA	
KO3	.98	.97	NA
	KO1	KO2	KO3

2) The authors need to prepare a Volcano plot for all identified peptides (ADMA modified and unmodified peptides), analogous to the data presented in Figure 1C. In this figure,

the authors can then specifically highlight the ADMA containing peptides, which should be significantly regulated as compared to unmodified counterparts. Otherwise the quantitative aspects of the experiment cannot be used to infer which proteins are CARM1-substrates in an unbiased manner.

Below are two volcano plots depicting changes in abundance of both ADMA-containing and unmodified peptides in MCF7 and MDA-MB-231 cells. As evident from the plots, a subset of modified peptides in both cell lines exhibited extreme reduction in abundance with very high significance, and no unmodified peptides behaved in the similar manner. Note, there are >65,000 unmodified peptides depicted in each plot, and only a very small number of them (<1%) crossed the threshold we imposed for categorization as a putative CARM1 substrate (greater than two-fold reduction with p-value less than 0.01; cutoffs are shown in dashed lines). Additionally, as CARM1 functions as a gene expression regulator, some changes in protein and therefore peptide abundance are expected. These plots were added to Supplementary Figure S1C.

3) Please prepare a heat map analysis for all unmodified peptides as well in a similar manner to the heat maps presented in Figure 1B, 4E and S1B.

Below are two heat maps depicting log₂ transformed mean normalized intensities of unmodified peptides in MCF7 and MDA-MB-231 cells. Although small clusters of changing peptides could be noticed, the changes were not as pronounced and/or consistent as they were for modified peptides in Figures 1B & 4E and Supplementary Figures S1B.

- From the experimental setup the authors used three replicates to obtain their results. However, the manuscript does not state whether these replicates are biological or technical replicates? Do the analyzed cells stem from three independent experiments, or do they stem from the same experiments being split into three?

We apologize for omitting this important information. All experiments were carried out using three biological replicates meaning three individual cell cultures were grown separately. These details were added to the Methods section, the main text of the manuscript, and the caption of Figure 1A. The text of the Materials and Methods section now includes the following sentence “Cells for all MS experiments were grown in biological triplicates as three individual cell cultures.” The main text says “To do this we immunoprecipitated ADMA-containing peptides using ADMA antibodies in two parental and CARM1 KO paired cell lines, MCF7 and MDA-MB-231, and mapped the sites of arginine methylation using nano-liquid chromatography tandem mass spectrometry (nanoLC-MS/MS) in biological triplicates (**Figure 1A**).” The caption of Figure 1A states

“Enrichment of asymmetric dimethylarginine (ADMA) - containing tryptic peptides from three biological replicas of parental and CARM1 KO cell lines was followed by quantitative mass spectrometry using tandem mass tags (TMT).”

- The authors only use FDR correction at the peptide level for ADMA peptides. FDR on all identified peptides should be included!

We agree and did process all peptides using 1% FDR correction; however, we failed to clearly convey that in the original draft. Briefly, the final list of ADMA-containing peptides was produced after separate FDR correction only using search results that corresponded to target and decoy peptides carrying the modification(s). This approach is more conservative and is strongly favored by other researchers studying arginine methylation via LC-MS/MS (Hart-Smith et al, MCP 2014; Larsen et al., Science Signaling 2016). Indeed, as the reviewer suggested, all identified peptides, both modified and unmodified ones, did undergo 1% FDR correction according to the accepted standards of proteomics data analysis (Elias & Gygi, Methods Mol Bio 2010).

We recognize that the original description of this aspect of the study was confusing, so we altered the text of the Methods section to clarify it. The modified sentences read *“1% false discovery rate (FDR) correction was performed on all identified peptides and proteins. To increase confidence in identified ADMA-containing peptides, separate 1% FDR was performed on search results that only corresponded to target and decoy peptides containing the modification.”*

- How clean are the ADMA enriched samples? How many percent of the total number of identified peptides in the experiments are actually identified as ADMA peptides? Please include such evaluation data.

The enrichment success varied between 12 and 26% that is similar or better comparing to a previously published study by Guo et al. (MCP 2014) using the same pan-specific ADMA antibody. We added this information to the Methods section of the manuscript that now states *“The enrichment success, i.e. the number of modified peptides among all peptides detected in the precipitate, varied from 12 to 26% across all experiments.”*

- The authors look into changes in overall protein abundance of the identified substrates. However, in this analysis a decrease in protein abundance is observed for several of the other PRMTs (especially since PRMT1 and 6 are both type I enzymes that catalyze ADMA). Please explain whether a decrease in ADMA levels could be connected to a decrease in expression of these type I PRMTs?

The abundance of PRMT6 decreased only by 1.6 fold in CARM1 KO MCF7 cells and remained virtually unchanged in CARM1 KO MDA-MB-231 cells. Among methylated peptides extracted from MCF7 cells we detected a peptide encompassing automethylation site of PRMT6 (Singhroy et al., Retrovirology 2013), and its abundance was unaffected by the knockout of CARM1. Meanwhile, no known substrates of PRMT6 were detected among peptides reduced in abundance. Thus, we believe that the observed minor change in PRMT6 abundance in MCF7 cells was not biologically

meaningful and likely did not contribute to the measured decrease in the abundance of various ADMA –containing peptides.

As we report in the manuscript, the abundance of PRMT1 decreased by 2.5 fold in CARM1 KO MCF7 cells as compared to parental MCF7 cells but not in MDA-MB-231 cells. However, similarly to PRMT6, this decrease likely had a limited effect on its catalytic activity as i) abundances of many known substrates of PRMT1 were unaffected in MCF7 cells (HNRNPA0, HNRNPA3, HNRNPAB, HNRNPH3, HNRNPU, HNRNPUL2, HNRNPK,

ALYREF, CHTOP, CAPRIN1, FUS, etc.); ii) no known substrates of PRMT1 were detected among peptides reduced in abundance; and iii) the GAR (RGG/RG) motif, well-known to be methylated by PRMT1, was not extracted from the proposed substrates of CARM1 but was detected among peptides whose abundance was unaffected by CARM1 knockout (Supplementary Figure S2A).

Lastly and most importantly, following suggestions of our reviewers, we re-designed a peptide array and performed *in vitro* methylation assay (panel to the left; Figure 2D in the manuscript) with purified PRMT1, 5 and PRMT6 including all appropriate controls. Low cross-activities were detected towards any of the tested putative CARM1 substrate sequences by PRMT1, 5 and PRMT6, while these enzymes were shown active towards

their respective known substrates in the same experimental setting. The results of these experiments strongly suggest that the reduced ADMA sites were primarily substrates of CARM1, not of other PRMTs.

- Besides, several of the known PRMTs are not identified in the MS analysis – please include a WB for all different PRMTs (at least for all type I PRMTs that preferentially catalyze ADMA) for WT and KO CARM1 cells to determine the expression levels of these in the experiment setup.

The abundance of all type I PRMTs (PRMT1, 3, 4, and 6) was quantified in the MS experiments in both cell lines. The only exception is PRMT8 that is believed to be a brain-specific enzyme (Lee et al., JBC 2005). Previous in-depth proteomic characterization of MCF7 cells (Geiger et al., MCP 2012) also did not identify PRMT8 among >10,000

proteins they detected. Neither did other in-depth studies of global protein expression in MCF7 (Nie et al., Proteomics 2015; Sacco et al., Cell Systems 2016) and MDA-MB-231 cells (Hoedt et al., PLOS One 2014; Lawrence et al., Cell Systems 2015). Thus, PRMT8 is unlikely to be expressed in either breast cancer cell line we used.

As requested by the reviewer, we performed Western blots, and their results are shown to the right and are included in the revised version of the manuscript (Supplementary Figure S1). In agreement with the MS data, none of the type I PRMTs measured by Western blotting exhibited strong changes in abundance upon knockout of CARM1, except PRMT1 in MCF7 cells, and the implication of this finding is discussed in the manuscript. Data for PRMT8 are not shown as we were unable to detect this enzyme in any cell line. We also observed modest reduction in the protein abundance of PRMT5 in CARM1 KO cells examined by Western blotting. However, this reduction is unlikely to affect the interpretation of our findings given that PRMT5 is a type II PRMT catalyzing SDMA.

- For the bioinformatic analyses presented in Figure 1D, Figure 2A and 2B the authors seem to be making an unintentional mistake in their comparison between ADMA containing arginines and unmodified arginine residues. All the presented data seems to stem from comparisons between identified ADMA sites and essentially remaining arginine residues in the human proteome. However, such an approach may unfortunately lead to false biases as the proteomics experiment certainly is abundance biased – i.e. any proteomics experiment has a strong tendency for identifying more abundant proteins as compared to low abundant ones. Consequently, the observed difference between ADMA sites and arginine residues (AnyR) may simply reflect a difference in the proteins being compared and not necessarily a difference related to ADMA. For example, the identified ADMA sites reside on proteins involved in RNA-associated processes, and these proteins have been described to harbor a larger number of somatic mutations as compared to other proteins. Hence, the observed difference showed in Figure 1D may simply be due to the ADMA sites primarily being identified on RNA-associated proteins – which their analysis strongly support.

To evaluate this, and remove any possible abundance or protein-group bias, the authors will have to redo their analyses and only compare arginine residues residing on the same proteins. That is, for a given protein X where an ADMA site has been identified the authors need to compare this ADMA site to a randomly chosen arginine in protein X that is not methylated. This is the only way how any abundance bias can be removed. Moreover, this way the authors also ensure that the two datasets being compared (ADMA sites versus nonmodified sites) actually contain that same number of sites.

We agree with the reviewer's assertion that proteomic experiments generally suffer from the abundance bias. However, we, and others, argue that analyses of post-translational modifications are less affected by this issue, as the immunoprecipitation and other methods of PTM enrichment rely on the abundance of the PTM itself, not the protein that bears it, i.e., a strongly modified low abundance protein is as likely to be pulled down as a lowly modified high abundance protein. Hundreds of quantitative proteomic manuscripts are published each year, and the suggested approach is not routinely implemented in them.

Still we investigated the issue of abundance bias by obtaining copy number per cell estimates, generated using a variety of human cell lines (Wisniewski et al., MCP 2014). We used them to calculate absolute abundances of the detected CARM1 substrates and compared them to the general human proteome. The plot on the left demonstrates that

the distribution of the abundances of CARM1 substrates (in black) closely resembled that of all human proteins (in red). On average, the detected CARM1 substrates may be even less abundant than all human proteins in general, as suggested by the slightly smaller median of this population (4.85 and 4.92, respectively). Thus the list of proteins we report as CARM1 substrates is unlikely to be skewed towards more highly abundant proteins, and the analyses using it should not suffer from the abundance bias.

The RNA-associated proteins that are more likely to harbor somatic mutations (Larsen et al., Science Signaling 2016) are mostly heterogeneous nuclear ribonucleoproteins (HNRNPs). These highly abundant proteins are components of the spliceosomal machinery, and many of them are well-known substrates of PRMT1 and PRMT5 (HNRNPA0, HNRNPA3, HNRNPAB, HNRNPH3, HNRNPU, HNRNPUL2, etc.). In our dataset, abundances of all HNRNPs, but HNRNPM, were unaffected by CARM1 deletion and were not included in the list of proposed CARM1 substrates.

Further, methylation of low abundance substrates may be a general property of CARM1, as suggested by its relatively low K_m value. The K_m of CARM1 towards its natural histone substrate is only 0.15 μ M (Jacques et al., 2016) that is over twenty-fold less than the K_m of PRMT1 towards its histone substrate (4.2 μ M; Pak et al., Biochem 2011).

Thus although we do not disagree with the reviewer that proteomics experiments in general suffer from the abundance bias, given the special properties of our dataset and the general practice of the proteomics field, we believe this issue did not confound our bioinformatics analyses.

- Likewise, the analyses shown in Figure 2A and 2B needs to be done in a similar way as described above. Comparison of one ADMA site on protein x to a randomly chosen

unmodified arginine on the same protein x. Especially considering that the observations that the authors want to report may in fact just stem from the abovementioned abundance bias.

See the response above.

- For the *in vitro* experiments described in Figure 2C, 2D and 2E, the authors need to include proper controls. For example, not positive or negative control is included in the analyses for PRMT1, PRMT5 or PRMT6 shown in figure 2E. Hence, the observed difference between figure 2E and 2D may just simply stem for the various blots being exposed differently. The authors therefore need to include such controls analogous to those included in Figure 2C.

As suggested by the reviewer, we designed another 96-spot peptide array which includes 17 known substrates of PRMT1, 10 known substrates of PRMT5, and 4 known substrates of PRMT6 as positive controls of proper function of specific enzyme in the *in vitro* methylation assays. The results are now shown in the new Figure 2E and are included in the response above. Consistently with the *in vitro* methylation results obtained from the previous 192-spot peptide array, assays using the new peptide array confirmed the ability of CARM1 to methylate over 90% of the putative substrates detected by IP-MS.

As stated in our revised manuscript, “Assays using purified PRMT1, 5, and 6 (**Figure 2E**, as indicated) revealed that a few substrates could be methylated by both CARM1 and other PRMTs. Specifically, ~8%, ~5%, and ~19% of the tested CARM1 substrates were also methylated by PRMT1, PRMT5, or PRMT6, respectively. Concomitant control experiments with respective known substrates of these PRMTs validated their normal enzymatic activity; PRMT1 methylated 9 out of 17 tested substrates, PRMT5 - 2 out 10, and PRMT6 – 3 out 4. Overall, these data demonstrate that although some overlap in substrate preference occurred between different PRMTs, CARM1 was prevalently selective towards the identified peptide sequences designated as CARM1 substrates by the LC-MS/MS experiments.”

- Likewise, the authors will have to comment on why there are so many empty spots for the identified CARM1 substrates shown in Figure 2D? Identification of substrates via their MS experiment should entail a 1% FDR, but from figure 2D it seems that a large number of spots show no real coloring (i.e. does not constitute proper CARM1 substrates). And many of these empty spots even seem reproducible between the two replicates?

We aren't entirely sure what the reviewer refers to as empty spots. We acknowledge that the signal intensities for different substrates varied, and perhaps the reviewer might have recognized some substrates with weak autoradiography signal as empty spots in 192-spot array format. As our initial classification of putative CARM1 substrates (two-fold reduction at p value > 0.01) was somewhat rough, we would not expect exactly 1% FDR among the selected substrates. Additionally, methylation of some substrates may require the presence of intra-cellular factors that are absent in the *in vitro* assay and therefore result in some false negatives.

As shown in Figure 2D, 90% of the putative substrate sequences (identified from the MS experiment) could be methylated by CARM1 *in vitro*. Methylation of peptides in new 96-spot peptide array format is now also presented to further clarify this issue.

Minor points:

- Please include a rationale why the MS experiments were not performed using MS3 to avoid ratio compression? Especially considering that the data was measured on an Orbitrap Lumos Fusion instrument which is fully capable of analyzing MS3?

We didn't include MS3 scans because they necessitate a large number of ions and long injection times, and the low abundance of ADMA-containing species did not readily permit these conditions. As we mostly classified sites in a binary fashion (reduced or non-reduced), being able to quantify the precise fold of reduction was less valuable than the ability to successfully sequence more peptides, which would have been compromised by the addition of MS3 scans.

- Please include resolution used for MS/MS scans.

We used the resolving power of 60,000 at m/z 200. This information was added to the Methods section that now states "*Tandem MS scans were collected in the Orbitrap at a resolving power of 60,000 at 200 m/z on precursors with 2-8 charge states, using HCD fragmentation with normalized collision energy of 35 and dynamic exclusion of 100 s.*"

- The collision energy used seems rather high. Any reasons for this?

Our prior studies have found that normalized collision energy of 35% is necessary for efficient cleavage of the TMT reporter region and therefore, quantification. Other laboratories also routinely use fragmentation energy of 32% or higher (Dillon et al., JPR 2011; McAlister et al., Anal Chem 2012; Wuhr et al., Anal Chem 2012).

- In the methods section the authors describe that they used TMT10-plex, but the data described in figure one is only TMT6-plex?

As the reviewer correctly points out, we used six labels for the two experiments described in Figure 1 but nine labels for the experiment depicted in Figure 4. TMT 6-plex kit is contained within TMT 10-plex, therefore we named TMT 10-plex as the product we used. To avoid further confusion, we have added a table to the Materials and Methods section (Table MM1; please see below in the response to Reviewer #3) listing all experiments, replicates, TMT kits, and respective TMT channels used.

- On page 2 the authors claim that mono-methylation is only catalyzed by PRMT7. However, it is widely known that the majority of PRMT enzymes can catalyze mono-methylation (Bedford and Clarke, 2009), including CARM1. However, certain PRMT enzymes primarily catalyze di-methylation, but has still been reported to catalyze mono as well.

To avoid any confusion, we added the word “final” to the sentence the reviewer refers to. Now the sentence reads “*Via transferring a methyl group from S-adenosyl-L-methionine (SAM) to the side chain of arginine residues, PRMTs catalyze formation of three final product types: ω -N^G-monomethylated arginine (MMA) by PRMT7, ω -N^G,N^G-asymmetric dimethylarginine (ADMA) by Type I PRMTs, and ω -N^G,N^G-symmetric dimethylarginine (SDMA) by Type II PRMTs.*”

- The authors use both MCF7 and the triple negative cell line MDA-MB-231. They find less CARM1 substrates in MDA-MB-231 cells, but they do not at any point comment on possible explanations for this finding. (And they could maybe state that only the MCF7 cells are used for the rest of the experiments and why this is).

We speculate that the varying enrichment success and/or the existence of fewer CARM1 substrates in MDA-MB-231 cells could explain this difference. A recent microarray study of CARM1 across various breast tissue types reveals its diverse effects on gene expression (Mann et al., Carcinogenesis 2013). It's reasonable to suggest that its enzymatic activity also varies in different tissue/cell types.

We added the following sentence discussing this issue to the main text of the manuscript “*Fewer ADMA-containing peptides were detected in MDA-MB-231 cells as compared to MCF7 cells, likely due to differential substrate expression and/or CARM1 activity between two cell lines, as well as variable enrichment success.*”

- Figure 3C; please include a loading control.

The same loading control was shared between the experiments in Figure 3B & C. We apologize for omitting this information, and the modified legend of the figure now reads “*The corresponding loading control was shared between the experiments in panel B (BAF155, MED12, and PABP1) and C and is depicted in panel B labelled with IB-FLAG.*”

- Figure 4E; there seems to be an increase in abundance of some ADMA peptides for both CARM1 140-608 cells and CARM1 KO cells (maybe even more in the CARM1 140-608 cells compared to the KO cells). The authors mention in the main text that the observed increase in abundance could be due to a decrease in PRMT1. However, at the same time they state that no decrease in PRMT1 abundance is observed in CARM1 140-608 cells. Please explain? Could the observed difference simply constitute an artifact of using z-scoring for the generation of heat-maps?

The heat map on Figure 4E displays log₂ transformed ADMA peptide intensities in CARM1 KO cells and cells expressing truncated CARM1. These intensities are normalized to the intensities of the respective peptides in cells expressing full length CARM1 and are not z-score transformed. Therefore, the observed increase in abundance is unlikely an artifact of data processing.

While the observation of PRMT1 decrease in CARM1 KO but not CARM1 140-608 cells is interesting, we do not know the exact reason account for the difference. One possible

explanation is that N-terminal of CARM1 regulates PRMT1 stability, but we do not have further evidence in support of this speculation.

- Figure 1F: How does the Pearson correlation look like when comparing CARM1 140-608 to wild-type? This correlation should be similar to the one that the authors need to include for each of their initial WT vs KO experiments (as mentioned in the first point of these comments).

We compared \log_2 transformed intensities of unmodified and modified peptides, averaged across three replicas of CARM1 28-608 and CARM1 140-608 samples (see the plots below). As evident from the plots and discussed in the manuscript, the abundance of unmodified peptides was largely unaffected by the N-terminus truncation, as the CARM1's ability to regulate gene expression mostly stems from its function as a cofactor, not a methyltransferase. However, abundances of ADMA modified peptides vastly differed between the cells expressing the full length enzyme and its N-terminus truncated version, as demonstrated by a considerably lower R^2 value (.99 and .65 for unmodified and modified peptides, respectively). These data support our major conclusion that the N-terminus of CARM1 is essential for its enzymatic activity. The plots below were added to Supplementary Figure S4F.

Reviewer #3 (Remarks to the Author):

In the current manuscript the authors use a quantitative MS-based proteomics approach to carry out an unbiased de novo and peptide-specific identification of CARM1 substrates in breast cancer cells. This led to the identification of 130 putative CARM1 protein substrates, the vast majority of which was validated in vitro through an ad hoc built peptide-array whereby a list of 200 peptides (among experimentally ID e predicted ones) was tested as substrates for enzymatic methylation assays using recombinant CARM1 and other PRMTs as control to confirm specificity. Additionally, bioinformatics analysis of the newly identified sites allow to definition of a putative novel CRM1 recognition motive,

whereby the enrichment of Proline in the vicinity of methylated Arginine is confirmed whereas Glycine and Methionine not. In the second part of the paper the authors focus on analyzing the functional role of the N-terminal domain of CARM1 and with a set of biochemical in vitro assays demonstrate that it is necessary for the efficient recognition, binding and methylation of the Proline/Arginine -rich motifs. Hence, this study extends the knowledge about the activity and substrate recognition of this important enzyme (which is aberrantly expressed in many cancers and is a promising target for therapy through the development of inhibitors), thus providing a useful resource for the design of novel, more specific inhibitors that may take into account the newly discovered N-terminal domain function.

The topic addressed by this study is undoubtedly relevant, interesting and very hot and the MS-proteomics- approach may in principle extend significantly the knowledge on the molecular activity/specificity of CARM1 protein-methyl-transferase, an enzyme aberrantly expressed in many cancer types. It is overall well designed and well written, although the two parts composing the paper (the MS-proteomics dissection of the CARM1- substrates and the biochemical analysis of the function of CARM1 N-terminal domain) appear to be quite independent/disjointed/separated.

We are grateful to the reviewer for the expressed confidence in the value of our work and the thoughtful critiques provided below.

In spite of its scientific relevance and novelty and the overall innovative approach, the paper suffers from some major (and minor) limitation that must be addressed to improve quality, reliability of the data and the overall impact of the study. Such issues are outlined in the points below.

Major issues:

1) The major novelty of the paper lays on the of unbiased identification of about 300 novel CARM1 –target sites on almost 140 proteins through a MS-based proteomics approach based on sample fractionation, asymmetric di-methylated peptide enrichment by immunoaffinity (anti-pan-ADAM antibody from CST) and TMT-based quantitative analysis of methyl-variations upon CARM1 KO. The approach followed is based very recent technical advancements in the field of the methyl-proteome analysis by MS. However, precise details about the MS-data and their analysis, which led to the identification of the 300 sites, are totally absent. I am aware that Nat Comm. is NOT a proteomics journal; nevertheless, because the major novelty of the manuscript resides in the collection of these new sites, it is mandatory to provide (as supplemental or through a repository) all MS/MS fragmentation spectra for the newly identified methylated peptides, to allow scientists judging the reliability of the information extrapolated. This is especially relevant since the authors state that they have carried out the validation of the novel asymmetric-di-methylated sites peptide through the manual annotation of the fragment peptides and the manual identification of the marker ion derived from the neutral loss of DMA. Provide the MS/MS fragmentation spectra for at least for a reasonable subset (at least 30%) of these peptide peptides is therefore essential. Fragment ions and peptide coverage should

be clearly displayed to demonstrate both that the methylation is precisely assigned and that the specific report ion for ADMA is always detectable, as stated.

Indeed the neutral loss ion fragment, which the authors refer to, is described in C. J. Brame et al , where the authors state that: “We speculate these ions arise from neutral loss of monomethylamine, dimethylcarbodiimide, and dimethylamine”... Actually, these are not very well-established diagnostic ions for MMA/ADMA by the community and have not been extensively employed by other groups working on methyl-proteomes to discern between ADMA/SDMA. Hence, the authors should provide not only the manually annotated MS/MS spectra containing these diagnostic ions, but also many more details on the analysis of MS data in Experimental section.

We completely agree with this critique and made multiple improvements to accommodate the reviewer’s request. First of all, we deposited all raw data files into PRIDE (Accession #PXD005871), and the reviewer can use the following information to download the data for further examination:

Username: reviewer25310@ebi.ac.uk

Password: eCZI56LQ

Originally the data were deposited into Chorus (ID 1174), another public proteomics data depository, and we would be happy to share a permission to download data though this resource if preferred.

Second, we annotated ALL spectra corresponding to ADMA-containing peptides that are available for download along with the raw data in PRIDE (Accession #PXD005871). On average, we achieved sequence coverage of 71.5% (+/- 21.7%) and 75.2% (+/- 21.2%) without and with neutral loss consideration, respectively. Given that that the average peptide length in this dataset is fairly large - 26 amino acids, the observed sequence coverage was quite high and gives us confidence that the ADMA-containing peptides were identified correctly. The increase in sequence coverage upon consideration of fragments resulting from the loss of dimethylamine (~4%) indicates that the neutral loss occurred consistently throughout the experiments and could benefit assignment of generated fragments.

Lastly, we greatly expanded the description of experimental details, specifically search parameters, reporter ion quantification, spectrum annotation, and neutral loss assignment. We also included a table listing all samples and corresponding TMT channels used in each experiment (Table MM1; see below in this response for more information).

The following is the extended explanation in the Materials and Methods section of the manuscript: “*Generated spectra were searched against the reviewed Uniprot database of human protein isoforms (downloaded 1.19.2015) and processed using the COMPASS software suite (2). Carbamidomethylation (+57.0513 Da) of cysteine residues and TMT 10plex (+229.1629 Da) on N-termini of proteins and lysine residues were included as fixed modifications. Oxidation of methionine (+15.999 Da), TMT 10plex on tyrosine*

(+229.1629 Da), and di-methylation of arginine (+28.0313 Da) were included as variable modifications. Average mass tolerances of 125 ppm and 0.015 Da were allowed for MS1 precursor searches and MS2 fragment searches, respectively. Up to 3 missed cleavages on tryptic peptides with proline rule were allowed. 1% false discovery rate (FDR) correction was performed on all identified peptides and proteins. To increase confidence in identified ADMA-containing peptides, separate 1% FDR was performed on search results that corresponded to target and decoy peptides containing the modification. TMT reporter region quantification was performed using an in-house software TagQuant, as previously described (3). Briefly, the raw reporter ion intensity in each TMT channel was corrected for isotope impurities, as specified by the manufacturer for the used product lot, and normalized for mixing differences by equalizing the total signal in each channel. In cases where no signal was detected in a channel, the missing value was assigned with the noise level of the original spectrum (i.e. noise-band capping of missing channels), and the resultant intensity was not corrected for impurities or normalized for uneven mixing. All spectra corresponding to ADMA-containing peptides were computationally annotated using in-house software Annotated Spectrum Generator. Briefly, each peptide was fragmented in silico into its b- and y-type product ions. Using exact mass and an allowed ± 10 ppm tolerance, each fragment was searched for in the associated MS² spectrum, and matched peaks were annotated with product ion type and number. All m/z peaks having an intensity < 1% of the MS² base peak were eliminated from consideration. All charge states less than the parent precursor charge state were considered when matching fragments to m/z peaks in the MS² spectrum. Ions produced upon the characteristic neural loss of dimethylamine (-45.0837 Da) were identified by subtracting the lost mass off the mass of arginines carrying ADMA modification and allowing for charge state changes.”

2) The second major limitation concerns to the experimental evidence of Suppl. Figure S1C, that PRMT1 is down-regulated 2.5 fold by CARM1 KO in MCF7 cells. This is a very major problem that could jeopardize the analysis, given that PRMT-1 is the major type -I PRMT, accountable for more than 80% of the R-methyl-proteome. A 2.5 fold decrease of this enzyme is a very significant change and must have a remarkable impact on the set of the asymmetric-di-methylated peptides investigated in this study. By no means imposing a 2-fold change cut-off in the methyl-peptide abundance can provide a solution for this specificity problem as proposed by the authors. This is a quantitative filtering criterion that does not distinguish and discriminate for substrate specificity! Since PRMT1 down-regulation appears to be cell-type dependent -and in fact it does not occur in MDA-MB-231 cells- the authors should r-select breast cancer cell lines that do not display PRMT1 down regulation upon CARM1 KO, like the MDA-MB-213 cell line and exclude MCF7 from the proteomic screening. On this line, remarkably the authors do not specify whether their list of 300 methylated peptides is derived from the INTERSECTION or the UNION of the two screenings made on MCF7 and MDA-MB-231, respectively: while the intersection (all peptide are methylated and down regulated >2 fold upon KO in BOTH

cell lines) would be acceptable, the UNION is not good and the peptide that are downregulated in MCF7 only are potentially false positive and should be excluded.

As we report in the manuscript, the abundance of PRMT1 decreased by 2.5 fold in CARM1 KO MCF7 cells as compared to CARM1 WT MCF7 cells but not in MDA-MB-231 cells. However, this decrease likely had a limited effect on its catalytic activity as i)

abundances of many known substrates of PRMT1 were unaffected in MCF7 cells (HNRNPA0, HNRNPA3, HNRNPAB, HNRNPH3, HNRNPU, HNRNPUL2, HNRNPK, ALYREF, CHTOP, CAPRIN1, FUS, etc.); ii) no known substrates of PRMT1 were detected among peptides reduced in abundance; and iii) the GAR (RGG/RG) motif, well-known to be methylated by PRMT1, was not extracted from the proposed substrates of CARM1 but was detected among peptides whose abundance was unaffected by CARM1 knockout. Additionally, following suggestions of our reviewers, we performed another *in vitro* methylation assay (panel to the left; Figure 2 in the manuscript) with purified PRMTs including all appropriate controls and detected very little catalytic activity towards any of

the tested putative CARM1 substrate sequences by PRMT1.

Based on the evidence listed above, we do not agree with the reviewer that “*the UNION is not good and the peptide that are downregulated in MCF7 only are potentially false positive and should be excluded*”. Because fewer ADMA-containing peptides were detected in MDA-MB-231 cells as compared to MCF7 cells, the relatively lower number of substrates identified in MDA-MB-231 cells will restrain us from deriving a complete set of CARM1 substrates. As we validated most of the identified substrates by *in vitro* methylation assays using peptide arrays (Figure 2D and 2E), we are confident that the methylation sites detected in MCF7 cells only and in both cell lines were correctly identified as novel CARM1 substrates.

3) Strictly related to this is the remark made by the authors in the last paragraph of the results, when they comment on the up regulation of a minor subset of asymmetric dimethylated peptides upon CARM1 KO and CARM1 N-terminal deletion mutant and propose the explanation that PRMT1 is also down regulated and hence substrate scavenging activity by other PRMTs could take place on the R-sites set free from depleted PRMT1 (as described in Dhar. et al.). This hypothesis challenges the whole proteomics screening in Figure 1, because it implies that PRMT1 downregulation does have a

relevant effect of the methyl-proteome and that such effect may be composite and not easy to be dissected. In fact it can lead to either non compensated down-regulation of ADMA-peptides (which can be misinterpreted as false positive CARM1 substrates –as discussed), or effect which will be compensated and thus non-changing and non detectable, leading to false negatives. In any case, this highlights the limit of the current study and corroborates the need to carry out these proteomics –based screening for CARM1 peptides in model systems where not other PRMTs are affected.

Please see the response above.

4) With this high risk of false positive in the putative list of novel CARM1 target selected through the -2fold change cutoff, the *in vitro* validation experiment using the selected peptide array (Figure 2C-E) becomes vital to validate the MS data. Although this could be in principle a well designed assay, it misses important controls namely positive control peptides for PRMT1, PRMT5 and PRMT6 activity, to be included within the same array in order to clearly demonstrate that the enzymatic activity of these other PRMTs is comparable to that of CARM1. The results displayed in figure 2E are not convincing since it is not possible to: a) confirm the activity of the enzymes b) compare it directly with that of CARM1 within the same array, on a common set of substrates. This experiment, must then be repeated, in a modified form, to include these controls.

*As suggested by the reviewer, we designed another 96-spot peptide array which included 17 known substrates of PRMT1, 10 known substrates of PRMT5, and 4 known substrates of PRMT6 as positive controls of each specific enzyme in the *in vitro* methylation assays. The results are shown in the new Figure 2E and just above in this response. Consistently with the results from the previous 192-spot peptide array, assays using the new peptide array also confirmed the ability of CARM1 to methylate over 90% of the putative substrates. As stated in our revised manuscript, “Assays using purified PRMT1, 5, and 6 (Figure 2E, as indicated) revealed that a few substrates could be methylated by both CARM1 and other PRMTs. Specifically, ~8%, ~5%, and ~19% of the tested CARM1 substrates were also methylated by PRMT1, PRMT5, or PRMT6, respectively. Concomitant control experiments with respective known substrates of these PRMTs validated their normal enzymatic activity; PRMT1 methylated 9 out of 17 tested substrates, PRMT5 - 2 out 10, and PRMT6 – 3 out 4. Overall, these data demonstrate that although some overlap in substrate preference occurred between different PRMTs, CARM1 was prevalently selective towards the identified peptide sequences designated as CARM1 substrates by the LS/MS/MS experiments.”*

5) In the second part of the manuscript (figures 3 and 4 and associated supplementary data), all *in vitro* methylation experiments are carried out with BAF155, MED12, PABP1 as substrates. These are already known CARM1 substrates, well characterized in previous publications by the authors. It would be nice, (also with the aim of better linking the two parts of the story), to include in the same experiments a few of the substrates newly identified through the MS-screening. Additionally, it is unclear why in this part HEK293T wt and KO cells were used, given that the initial screening of CARM1 target

had been carried out in breast cancer cell lines. This contributes to make the two part of the manuscript even more disconnected/disjointed.

We believe that the two parts of the manuscript are quite coherent, and the experiments described in the second part constitute a logical progression of the findings reported in the first part of the study. First, we discovered novel CARM1 substrates by IP-MS, validated them using *in vitro* methylation assays, and established that CARM1 substrates contain proline-rich motifs (Figure 1 and 2). With the knowledge that the structure of the CARM1 N-terminal domain resembles those of EVH1 domains - a subfamily of PH domains that bind proline-rich sequences - we hypothesized that this domain of CARM1 is required for substrate recognition and methylation. The second part of the manuscript specifically tested this suggestion and found it to be correct (Figure 3 and 4). Therefore, the transition from the first part of the manuscript to the second one was perfectly logical and driven by a clear scientific hypothesis.

We took the reviewer's suggestion and included two more substrates for validation. TET2 is a new substrate identified in this study, and NCOA3 was previously discovered by another lab (Feng et al., MCB 2006). We validated TET2 as bona fide substrate of CARM1 using Western blotting and peptide arrays (New Figure 2C, D & E) and confirmed that the N-terminal domain of CARM1 is required for binding of both TET2 and NCOA3 (the panel to the left and New Figure 3B). Additionally, although BAF155, MED12, and PABP1 are known substrates, the process of their recognition by CARM1 was unknown at the time. In Figure 3 and 4, we showed that the N-terminal domain

of CARM1 is instrumental in recognition of these well-established CARM1 substrates – a valuable finding in its own right.

We used HEK293T cells in some experiments (Figure 3B & C), instead of the breast cancer cell lines, because HEK293T cells can be transfected with desired constructs at higher efficiency. Later in the study we stably expressed various CARM1 constructs in MCF7 cells (Figure 4) and resumed the experiments in this background.

Minor Points:

1) Since HpH based-fractionation before immuno-enrichment of methylated peptides has been shown to highly increase the detection of methyl-peptides (Ref 2), it is unclear why the authors chose to fractionate just the unmodified peptides fraction through this method.

Indeed, it would be much more worth fractionating the cellular peptidome before the methyl-peptides immuno-enrichment with pan-ADMA-peptides, as described in recent publications.

We agree that fractionation followed by multiple rounds of enrichment would have likely improved our results. Based on our work and studies by other laboratories (Ref. 2 and Guo et al., MCP 2014), efficient enrichment of ADMA peptides necessities over 10 mg of starting material, and unfortunately, our laboratory does not possess the capacity to fractionate such large amounts of peptides.

2) The figure 1B shows the unnormalized log₂ TMT ratios of the identified methyl-peptides and the corresponding protein ratio are displayed in a separate figure (S1C). However, it is advisable to normalize the TMT ratios of each modified peptides to the respective protein level and then display and analyze this normalized/corrected ratio, which would take into account for each modified peptide even any minor contribution of the respective protein abundance.

Following the reviewer's suggestion, we adjusted TMT ratios of ADMA peptides by the corresponding protein ratios, and the comparison between the original plot (Figure 1C, MCF7 cells) and the modified one is displayed below. Briefly, we mean normalized ADMA peptide intensities of all samples within the respective TMT experiment, repeated the same procedure on total protein abundance measurements, and subtracted the latter from the former. Note, there are fewer points on the right graph, as the list of proteins we quantified total abundances of did not include some of the substrates in Figure 1C (~10% of all ADMA-bearing proteins).

Overall, the adjustment slightly shifted the relative positions of individual data points, but it did not exclude any peptides classified as putative CARM1 substrates from this category. The only exceptions were the few proteins whose abundance decreased in our data set (Supplementary Figure S1D), which we had already excluded, as described in the manuscript.

3) In Figure 1B and S1B, the authors show that 50% of the identified methyl-peptides are regulated in CARM1 KO MCF7 cells and MDA-MB-231, respectively. However, in suppl. Table S1 several methyl-peptides contain more than one “methylatable” site, making the assignment of the observed regulation at the single site level difficult. Although we are aware that the site-specific assignment is challenging, this is an important point that may affect the motif-analysis and should be addressed and discussed by the authors.

The reviewer draws attention to the challenging issue that we have also pondered upon in the course of this study. As we measure changes in abundance of peptides, not specific sites, for peptides with multiple modifications we cannot confidently pinpoint methylation of which arginine residues is reduced.

To insure that the results of our motif analyses (Figures 2B and S2A) were not confounded by this discrepancy, we performed a control analysis extracting motifs from differentially reduced (top logo below) and unaffected (bottom logos) peptides that contained a single ADMA site. The extracted motifs in both cases were the same as the motifs in Figures 2B and S2A, although fewer different motifs were detected, perhaps due to the smaller number of submitted sequences and therefore decreased statistical power. Further, the percent of all submitted sequences that contained these motifs were very similar between the two analyses using only singly methylated and multiply methylated peptides (40.4% and 40.2%, respectively, for PR motif; 23.3% and 21% for RGG motif; 25.9% and 29% for RG motif).

The logos above were added to Supplementary Figure S2B & C, accompanied by the following text in the main body of the manuscript “As our method did not permit direct measurements of abundance changes on individual ADMA sites in the context of multiply di-methylated peptides, we performed control analyses by separately extracting motifs from singly di-methylated peptides. The detected motifs (**Supplementary Figure S2C & D**) were identical with several motifs on **Figure 2B** and **Supplementary Figure S2A**, therefore further corroborating their enrichment.”

Known CARM1 methylation sites on MED12 (Wang et al., Science Advances 2015; Gayatri et al., Scientific Reports 2016) and NCOA3 (Naeem et al., MCB 2007; Guo et al., MCP 2014) occur in close proximity to each other. We speculate that methylating several adjacent arginine residues may be a general characteristic of CARM1 methylation that contributes to the observed reproducibility of motif analysis between singly and multiply methylated peptides.

4) In general, figures containing the MS workflow, the MS spectra and the clustering analysis of TMT-labelled methyl peptides are too small and not really readable.

We have modified the figures and enlarged the text to make it easier to read.

5) The MS analysis is overall good quality, in spite of the limitation outlined at point 1. However, as said, the description of the Experimental procedures must be extended, with more information on the TMT quantification (e.g. the TMT labels used, the number of replicates, the experiment setup), the report ion analysis and detection from the neutral loss of ADMA.

We added a new table to the Materials and Methods section (Table MM1) listing all experiments, included replicates, and respective TMT labels.

Table MM1. Experimental setups and used TMT channels.

Experiment	Biological replicate	TMT channel
#1 (6-plex): wild type (WT) and CARM1 knockout (KO) MCF7 cells Figure 1B & C and S1C	WT1	126C
	WT2	127C
	WT3	128C
	KO1	129C
	KO2	130C
	KO3	131N
#2 (6-plex): wild type (WT) and CARM1 knockout (KO) MDA-MB-231 cells Figure 1C and S1B & C	WT1	126C
	WT2	127N
	WT3	127C
	KO1	128N
	KO2	128C
	KO3	129N
#3 (9-plex): knock-in wild type CARM1 (WT), knock-in CARM1 140-608 (TR), and CARM1 knockout (KO) MCF7 cells Figure 4E & F and S5	WT1	126C
	WT2	127N
	WT3	127C
	TR1	128C
	TR2	129N
	TR3	129C
	KO1	130N
	KO2	130C
	KO3	131N

We also extended the description of the report ion quantification in the Materials and Methods part of the manuscript. The added section is the following: “performed on search results that corresponded to target and decoy peptides containing the modification. TMT

reporter region quantification was performed using an in-house software TagQuant, as previously described (3). Briefly, the raw reporter ion intensity in each TMT channel was corrected for isotope impurities, as specified by the manufacturer for the used product lot, and normalized for mixing differences by equalizing the total signal in each channel. In cases where no signal was detected in a channel, the missing value was assigned with the noise level of the original spectrum (i.e. noise-band capping of missing channels), and the resultant intensity was not corrected for impurities or normalized for uneven mixing. All spectra corresponding to ADMA-containing peptides were computationally annotated using in-house software Annotated Spectrum Generator. Briefly, each peptide was fragmented in silico into its b- and y-type product ions. Using exact mass and an allowed ± 10 ppm tolerance, each fragment was searched for in the associated MS² spectrum, and matched peaks were annotated with product ion type and number. All m/z peaks having an intensity < 1% of the MS² base peak were eliminated from consideration. All charge states less than the parent precursor charge state were considered when matching fragments to m/z peaks in the MS² spectrum. Ions produced upon the characteristic neural loss of dimethylamine (-45.0837 Da) were identified by subtracting the lost mass off the mass of arginine residues carrying ADMA modification and allowing for charge state changes.”

6) Wrong numbering of Figures S4 E and S4 E in the main text.

Thanks for this correction, we have fixed the numbering.

7) In the first paragraph of the results, the sentence “..Our experimental design capitalized on the multiplexing capabilities of tandem mass tags (TMT), a technique that allowed us to enrich and analyze ADMA-modified peptides from several samples simultaneously..” is not formally correct as it delivers the concept that the use of TMT-labeling serves for methyl-peptide enrichment and detection. The TMT indeed is only useful for the quantitative labeling, and in particular for its multiplexing capability, which the authors did not take advantage of. This sentence should be therefore corrected.

The sentence intended to convey the idea that using TMT allowed us to enrich and analyze wild type and knockout samples and their replicas at the same time. Performing enrichment on all samples at once was particularly important, as this step could be very variable, and the use of TMT assured a complete overlap between peptide identifications made across all samples. We corrected the sentence that now reads “*Our experimental design capitalized on the multiplexing capabilities of tandem mass tags (TMT)³¹, a technique that allowed us to simultaneously enrich and analyze ADMA-modified peptides from wild type and knockout samples, assuring a complete overlap of peptide identifications between the two groups.*”

8) Overall is quite amazing – to be honest and a bit surprising – that the quantitative comparison of the asymmetric-dimethyl-peptides changes upon CARM1 KO and CARM1

N-terminal (140-608) mutant shows that the mutant seem to have almost a “linear” effect on methylation, exactly intermediate between wt and KO (figures 4E and F). This kind of linear/dose-dependent effects -affecting similarly most R-sites profiled, on most of the substrates analyzed- is rather unusual in global-MS-modification studies, so it somehow hard to imagine. Although we obviously do not question the accuracy of TMT quantification, maybe the authors could argue more about this impressive linearity of response.

This nice correlation strongly supported our conclusion that the N-terminal domain of CARM1 is required for substrate recognition. A small fraction of the substrates can bypass this prerequisite, but the most of them cannot be methylated to the levels comparable to those in the parental cells in the absence of the N-terminal domain.

In light of all this considerations, we think that the manuscript has potential for publications in this journal, but only upon major revision that will address the points outlined.

Editor’s feedback

We hope you will find the referees' comments useful as you decide how to proceed. Should further experimental data or analysis allow you to address these criticisms, we would be happy to look at a substantially revised manuscript. However, please bear in mind that we will be reluctant to approach the referees again in the absence of major revisions. Any re-submission should allow reviewer access to the mass spectra, preferably through a repository.

We have substantially revised the manuscript, addressed all of reviewers’ concerns, performed additional experiments, and provided access to mass spectra through three complementary venues – deposition in Chorus, deposition in PRIDE, and annotation of all spectra corresponding to ADMA-containing peptides. We believe these revisions have greatly improved the manuscript, and now our work meets high publication standards of Nature Communications.

Reviewers' Comments:

Reviewer #2 (Remarks to the Author):

The authors have adequately addressed the comments put forward by the reviewer.

One minor comment related to Figure S1A:

The presented scatter-plots in Figure S1A seem to have been analyzed using either imputation or similar data processing steps. At the lower abundance axes the distribution of data points appear rather "artificial" and non-random. It would be ideal to redo these scatter-plots and avoid any of the data processing/imputation steps, in order to obtain a proper correlation of the data.

Reviewer #3 (Remarks to the Author):

The revised manuscript NCOMMS-16-23707A addresses the majority of the reviewers' comments from the first revision, with substantial improvements from the initial work. Therefore I judge that the revised version of the manuscript is now acceptable for publication.

We are delighted to learn that the manuscript has been accepted and thank our editor and the reviewers for their valuable input. Please find below our responses to the reviewers' final comments (in blue).

REVIEWERS' COMMENTS:

Reviewer #2 (Remarks to the Author):

The authors have adequately addressed the comments put forward by the reviewer.

One minor comment related to Figure S1A:

The presented scatter-plots in Figure S1A seem to have been analyzed using either imputation or similar data processing steps. At the lower abundance axes the distribution of data points appear rather "artificial" and non-random. It would be ideal to redo these scatter-plots and avoid any of the data processing/imputation steps, in order to obtain a proper correlation of the data.

The data points the reviewer refers to are indeed products of noise-band capping – the missing values that were imputed using the value of noise in the respective spectra. Following the reviewer's suggestion, we removed the imputed data points, and this indeed slightly improved our correlations with new median R^2 of 0.984.

Below is the revised version of Figure S1A.

Reviewer #3 (Remarks to the Author):

The revised manuscript NCOMMS-16-23707A addresses the majority of the reviewers' comments from the first revision, with substantial improvements from the initial work.

Therefore I judge that the revised version of the manuscript is now acceptable for publication.